DOI: 10.1038/s41467-017-01430-6　　**OPEN**

# An extracellular matrix-related prognostic and predictive indicator for early-stage non-small cell lung cancer

Su Bin LIM [1,2], Swee Jin TAN[3,9], Wan-Teck LIM[4,5,6] & Chwee Teck LIM[1,2,7,8]

The prognosis and prediction of adjuvant chemotherapy (ACT) response in early-stage non-small cell lung cancer (NSCLC) patients remain poor in this era of personalized medicine. We hypothesize that extracellular matrix (ECM)-associated components could be potential markers for better diagnosis and prognosis due to their differential expression in 1,943 primary NSCLC tumors as compared to 303 normal lung tissues. Here we develop a 29-gene ECM-related prognostic and predictive indicator (EPPI). We validate a robust performance of the EPPI risk scoring system in multiple independent data sets, comprising a total of 2,071 early-stage NSCLC tumors. Patients are stratified according to the universal cutoff score based on the EPPI when applied in the clinical setting; the low-risk group has significantly better survival outcome. The functional EPPI gene set represents a potential genomic tool to improve patient selection in early-stage NSCLC to further derive the best benefits of ACT and prevent unnecessary treatment or ACT-associated morbidity.

[1] NUS Graduate School for Integrative Sciences & Engineering (NGS), National University of Singapore, #05-01, 28 Medical Drive, Singapore 117456, Singapore. [2] Department of Biomedical Engineering, National University of Singapore, 4 Engineering Drive 3, Engineering Block 4, #04-08, Singapore 117583, Singapore. [3] Clearbridge Accelerator Pte Ltd, Singapore 118257, Singapore. [4] Division of Medical Oncology, National Cancer Centre Singapore, 11 Hospital Drive, Singapore 169610, Singapore. [5] Office of Clinical Sciences, Duke-NUS Medical School, 8 College Road, Singapore 169857, Singapore. [6] Institute of Molecular and Cell Biology, A*Star, 61 Biopolis Drive, Proteos, Singapore 138673, Singapore. [7] Mechanobiology Institute, National University of Singapore, #10-01, 5A Engineering Drive 1, Singapore 117411, Singapore. [8] Biomedical Institute for Global Health Research and Technology, National University of Singapore, #14-01, MD6, 14 Medical Drive, Singapore 117599, Singapore. [9] Present address: Sysmex Asia Pacific Pte Ltd, 9 Tampines Grande, #06-18, Singapore 528735, Singapore. Correspondence and requests for materials should be addressed to C.T.L. (email: ctlim@nus.edu.sg)

Tumor stage is currently the best-established prognostic factor of patient survival[1]. Aside from this stage classification, much effort has been devoted to determine single oncogenes or biomarkers. In particular, recent successes in oncogene-directed therapies have improved patient survival[2]. For example, Oncotype DX testing (Genomic Health Inc, Redwood City, CA) has allowed improved prediction of recurrence risk and allocation of chemotherapy to patients who would most likely to benefit in breast cancer[3]. In contrast, lung cancer has seen few medical advances. A 5-year survival remains at 50–60% for stage I, 30–40% for stage II, and 15–20% for stage III lung cancer[4]. In particular, non-small cell lung cancer (NSCLC) remains the leading cause of cancer-related fatalities[5]. Although advanced or metastatic (late-stage) NSCLC has routine testing for targets such as epidermal growth factor receptor (*EGFR*) mutations and anaplastic lymphoma kinase (*ALK*) rearrangements, there is still lack of validated genetic risk stratification score to select patients who may best benefit from adjuvant chemotherapy (ACT) among early-stage resected NSCLC patients[4]. Adjuvant studies that randomized *EGFR* mutant lung cancer to adjuvant *EGFR* tyrosine kinase inhibitors or chemotherapy have also demonstrated no survival benefits[6]. Immense efforts in uncovering potential multigene assays for molecular prognostic tests have recently resulted in two genetic tests being marketed as Pervenio Lung RS (Life Technologies, West Sacramento, CA, USA) and myPlan Lung Cancer (Myriad Genetics, Inc., Salt Lake City, UT, USA). Although both assays have been validated in some patient cohorts[7,8] using formalin-fixed and paraffin-embedded (FFPE) specimens, the success of a prospective randomized clinical trial is still warranted for their immediate clinical utility.

Bidirectional crosstalk between tumor cells and their surrounding stroma is critical in tumor growth and progression[9]. Whereas cancer has been previously examined for genetic or epigenetic mutations in epithelial cells, it is now clear that the extracellular cues in tumor microenvironment (TME) also regulate cancer development and metastasis[10]. The extracellular matrix (ECM) components, in particular, have been perceived as important regulators in cancer progression[11]. Stromal and epithelial cells in cancer microenvironment jointly disrupt them and

their dynamics has become a hallmark of cancer[12]. Abnormal ECM, such as disrupted organization and changes in essential composition or topography of the ECM, has been implicated in cancer initiation and metastasis by remodeling the behavior of stromal cells and promoting tumor-associated angiogenesis and inflammation[11]. Hence, understanding how tumor-derived ECM components manifest in the diseased state may help identify new prognostic or therapeutic targets.

Here, our developed bioinformatics pipeline for large scale meta-analysis revealed differential expressions and significant enrichment of ECM-associated components in 1,943 primary NSCLC tumors relative to 303 normal lung tissues. We thus hypothesized that ECM genes play a significant role in predicting metastasis, recurrence risk and survival. A 29-gene ECM-related prognostic and predictive indicator (EPPI)[13] was specifically constructed and its robust performance was validated in multiple independent data sets comprising 2,071 early-stage NSCLC patients. To facilitate its clinical utility, the universal cutoff EPPI risk score, which stratifies patients with better overall survival (OS) was identified. This was further validated in identifying patients who would benefit from ACT in an independent validation cohort.

## Results

**Construction of a 29-gene EPPI related to NSCLC.** The bioinformatics workflow of our integrative genomic analysis is illustrated in Fig. 1. Batch effects arising from 10 independent microarray data sets were successfully adjusted as validated using principal components analysis (PCA). The first two principal components of the ComBat-transformed data (discovery set) which capture the most variance are shown in Fig. 2a. The PCA plot of samples adjusted for batch effect showed a clear separation between 925 primary NSCLC tumors and 193 normal lung tissue samples (Fig. 2b). We identified a total of 103 differentially expressed genes (DEG) that met our stringent statistical threshold in NSCLC compared to normal phenotype (Supplementary Table 1). Figure 2c shows a volcano plot revealing 32 upregulated and 71 downregulated genes in NSCLC. To dispel the existing bias against feature selection methods based on ranking genes, the

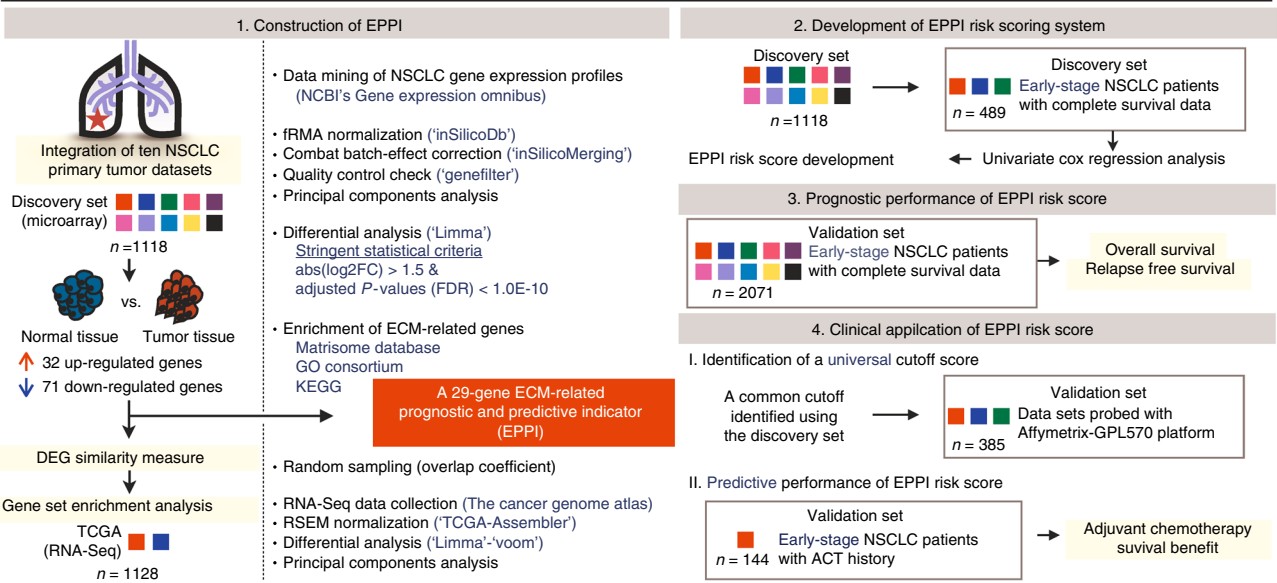

**Fig. 1** Schematic representation of the bioinformatics workflow. We constructed a 29-gene EPPI signature from 2,246 samples including primary NSCLC tumors and normal lung tissues using an integrative genomic approach. The EPPI risk scoring system was developed from discovery set comprising early-stage patients. The prognostic and predictive performance of the EPPI risk score was further validated in multiple independent validation sets comprising 2,071 early-stage NSCLC patients

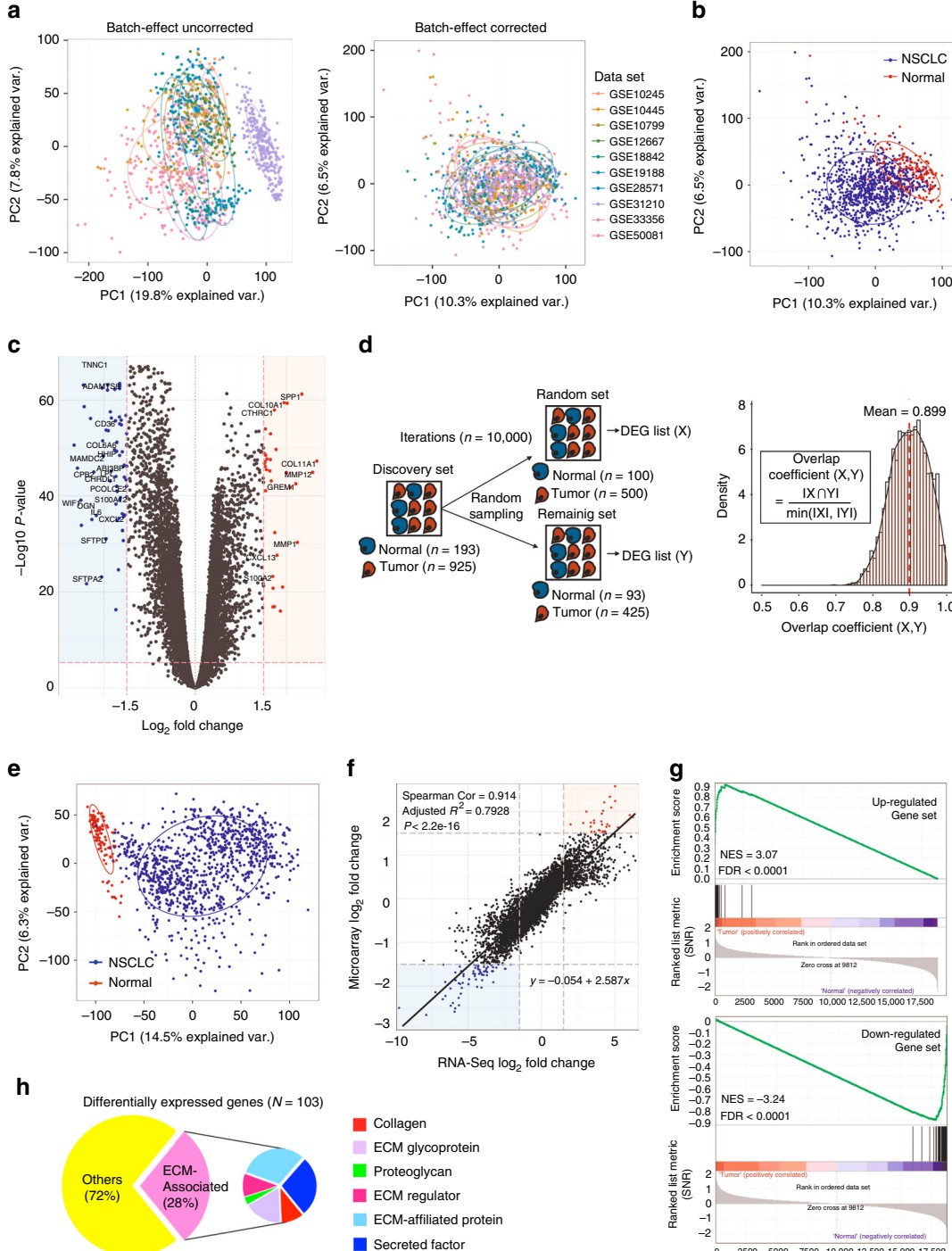

**Fig. 2** Construction of a 29-gene EPPI. **a** PCA plot of merged microarray data sets demonstrating the batch effect removal. Ellipses representing one standard deviation away from the mean of the Gaussian fitted to each data set. **b** PCA plot showing a clear separation between 925 primary tumors (blue) and 193 normal lung tissues (red). **c** Volcano plot of differential gene expression using merged microarray data. Upregulated and downregulated genes with absolute values of log2 fold-change >1.5 in primary NSCLC tumors are shown in red and blue, respectively. Gray dots represent statistically non-significant genes. **d** Schematic representation of random sampling steps for comparing the overlap between DEG lists using merged microarray data sets (left). Kernel density estimates of overlap coefficients computed from 10,000 iterations are shown (right). **e** PCA plot of TCGA data sets (LUAD, LUSC) showing a clear separation between primary 1,018 tumors (blue) and 110 normal lung tissues (red). **f** Comparison of log 2 fold-change estimate results from two different platforms. Upregulated and downregulated genes are shown in red and blue, respectively. **g** GSEA-generated enrichment plots of the upregulated and downregulated gene set in a TCGA cohort. The cumulative enrichment score is plotted as the green curve, which is the running sum of the weighted ES as the analysis walks down the ranked list computed from GSEA software. The vertical black lines on the middle portion of the plot indicate the position of inputted gene signatures in the ranked list of genes. Genes on the far left (red) and right (blue) correspond to high enrichment in NSCLC and normal phenotype, respectively. The bottom plot (gray) shows the value of the ranking metric as the computation goes down the list of ranked genes. The normalized enrichment score (NES) and the false discovery rate (FDR) are shown on the graph. **h** Pie chart of differentially expressed genes annotated with ECM-related molecules and their major categories for ECM constituents

overlap between DEG lists derived from random sampling was computed for 10,000 iterations (Fig. 2d). We took smaller subsets randomly from the discovery set and applied the same statistical criteria used to define the DEG list. The mean of the overlap coefficients computed for the iterations (mean overlap coefficient = 0.899) suggests there is significant overlap using the discovery set comprising 1,118 samples.

Since the DEGs used to construct the EPPI were identified from merged microarray data sets, we further validated the 103 DEGs in an independent cohort comprising RNA-Seq transcriptional profiling of 1,018 primary NSCLC tumors and 110 normal lung tissues from lung adenocarcinoma and squamous cell carcinoma patients (Fig. 2e). The estimates of log 2 fold changes of gene expression values between the samples across the two different platforms were highly correlated as shown in Fig. 2f (Spearman correlation = 0.914, $P < 2.2e\text{-}16$). Gene set enrichment analysis (GSEA)-generated results further showed significant enrichment scores for both upregulated (ES = 0.929, $P < 0.001$) and downregulated gene set (ES = −0.904, $P < 0.001$) in the diseased phenotype compared to normal tissues in the RNA-Seq platform (Fig. 2g).

Interestingly, gene ontology enrichment analysis of the 103 DEGs identified ECM-associated molecules, comprising 28% of the DEGs ($n = 29$), to be the most differentially enriched cellular components in the diseased state (Fig. 2h). We hypothesized that these identified ECM components play a significant role in the molecular pathways associated with NSCLC malignancy and their gene expression patterns may collectively predict clinical outcomes. Using three gene ontology databases, we constructed the EPPI gene set comprising the 29 ECM-related genes that were identified to be differentially expressed in primary NSCLC tumors (Supplementary Table 2).

**Personalized EPPI risk scoring system for patient classification.** We systematically analyzed the patient groups in the discovery data set to develop the EPPI based risk scoring system. Patients without complete survival information or at late-stage NSCLC were excluded in the univariate Cox regression analysis. Altogether, 489 early-stage NSCLC patients from the initial discovery set were included to generate the Cox regression coefficient of each EPPI gene (Supplementary Table 3). The EPPI risk score of each patient in the discovery set was then computed using the regression coefficient and represented collectively in the EPPI gene expression heatmap in Fig. 3a. The data are sorted in increasing EPPI risk scores and each patient's survival status is depicted as well. Forest plots illustrating the hazard ratios (HRs) and 95% confidence intervals (CIs) for each EPPI gene are presented with its respective Cox regression coefficient (Fig. 3b).

Time-dependent area under a receiver operating characteristic (ROC) curve (AUC) analyses were further performed in order to compare the prognostic performance of different patient classification methods. The AUC analyses for a 10-year survival demonstrated better performance of the EPPI risk scoring system compared to an unsupervised hierarchical clustering technique in classifying patients with different survival outcomes (Fig. 3c). The prognostic performance of different cutoff score was also compared in the AUC analyses to determine the best cutoff computation method. An optimal cutoff score, defined as the EPPI risk score with the most significant split using log-rank test, was selected to demonstrate the prognostic performance of the 29 EPPI genes in multiple independent data sets.

**Robust prognostic performance in early-stage tumors.** The ability of the 29-gene EPPI signature in stratifying early-stage NSCLC tumors with different prognosis was confirmed in

multiple validation cohorts comprising a total of 2071 early-stage NSCLC patients. Only stage IA to IIB patients with complete demographic information were included and patients were censored at their last examination, or at 10 years follow-up in all validation runs. The histology profiles of the samples included two studies comprising exclusively of lung adenocarcinomas (GSE68465, TCGA), and one of squamous cell carcinoma (GSE4573), and six of mixed histological types (GSE3141, GSE37745, GSE30219, GSE42127, GSE41271, and GSE11969). Patients annotated with ACT were excluded from the survival analyses.

Unsupervised hierarchical clustering of each validation cohort was first performed to demonstrate a robust prognostic performance of the EPPI gene signature in classifying patients with different survival outcomes (Supplementary Fig. 1). As demonstrated by consistently high HRs across validation cohorts, there was a significant association between survival and EPPI gene panel. We further tested the potential of the clinical utility of our developed EPPI risk scoring system in stratifying patients. The optimal cutoff value was determined for each cohort to divide early-stage patients into two classes, designated as low-risk and high-risk groups. Remarkably, the 29-gene EPPI was closely associated with OS in all validation cohorts (Fig. 4a). The predicted high-risk group had statistically significant shorter OS compared with those in the low-risk group as shown in univariate Cox regression analyses.

All validation cohorts demonstrated significant log-rank P-value ($P < 0.05$) despite the absence of few EPPI genes in some microarray platforms, demonstrating the strong prognostic performance of the EPPI gene signature in early-stage NSCLC patients. We also tested the association between the EPPI and RFS in the data sets using RFS as an endpoint (GSE50081 and GSE31210 from the discovery set, and GSE68465; Fig. 4b). The predicted high-risk group consistently had worse RFS outcomes than the low-risk group as shown in Fig. 4b.

Since only data sets with almost complete demographic information were evaluated, we further conducted multivariate Cox regression survival analyses to adjust for other clinical variables such as histology (ADC: adenocarcinoma; LCC: large cell carcinoma; SCC: squamous cell carcinoma), sex, smoking history, AJCC tumor staging (Stage IA–IIB) and age. Importantly, multivariate survival analyses showed that the EPPI signature remains to be an independent indicator of survival outcomes after the adjustment (Table 1). We further directly compared our model to these clinicopathological factors that are known to be associated with survival to some extent (Supplementary Fig. 3). Despite significant p-values for tumor staging, age and histology in few data sets, our gene signature has the strongest prognostic performance in most data sets (8 out of 10 validation cohorts). Although EPPI model also shows statistically significance in the other two data sets, histology and tumor staging shows better performance in GSE30219 and TCGA, respectively. Nevertheless, our gene panel remains to be the most significant factor compared to five clinicopathological factors in majority of early-stage NSCLC patients.

Gene clusters generated from hierarchical clustering using validation cohorts probed with the full-gene platforms (Discovery set, GSE3141, GSE37745, and GSE30219) revealed that high-risk groups associated with worse survival displayed consistently elevated expressions of collagens (COL10A1, COL11A1), matrix metallopeptidases (MMP1, MMP12), secreted factors (S100A2), glycoproteins (CTHRC1, SPP1), and ECM-affiliated proteins, or genes encoding proteins affiliated structurally or functionally to ECM proteins (GREM1) and low expressions of surfactant proteins (SFTPC, SFTPA2, SFTPD), secreted proteins (CHRDL1, WIFI), ECM-regulated genes (CPB2, MAMDC2, HHIP, LPL,

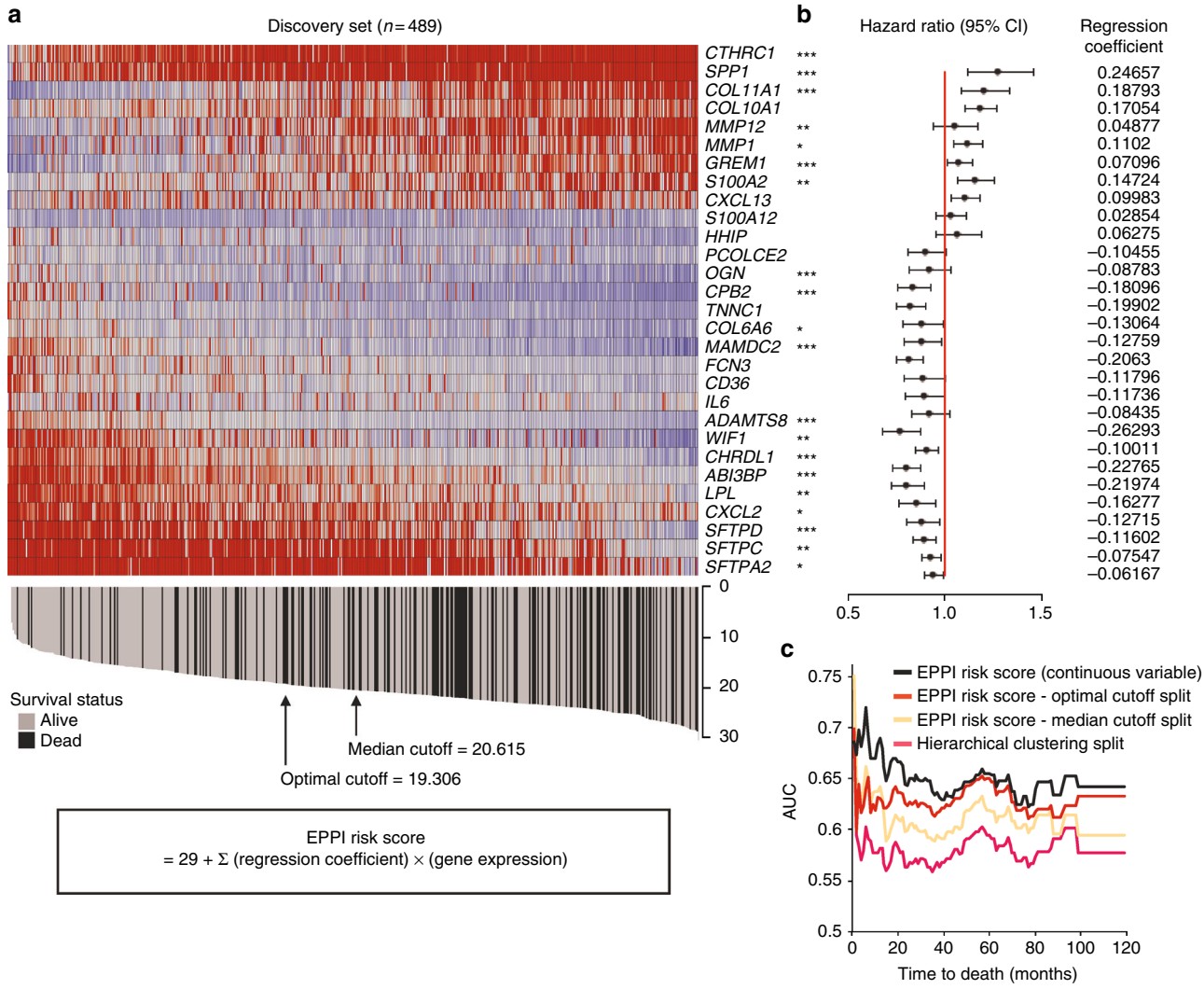

**Fig. 3** Development of the EPPI risk scoring system. **a** Personalized EPPI gene expression heatmap in the discovery set comprising only early-stage patients (red = high expression, blue = low expression). Columns are ordered by increasing EPPI risk score. The score distribution, survival status of the patient and the median/optimal cutoff scores are illustrated below. **b** Forest plots of each EPPI gene expression in the discovery set. The horizontal axis represent hazard ratio (HR) with 95% confidence intervals (CI) estimated using a Cox proportional hazards model. The asterisks represent the statistical significance in patient survival outcome (***$P < 0.001$, **$P < 0.01$, *$P < 0.05$; log-rank $P$-value). **c** Time-dependent area under ROC curves (AUC) for different patient classification methods were assessed in the discovery set

*CD36*, *ADAMTS8*), collagen (*COL6A6*), ECM-affiliated proteins (*FCN3*), ECM glycoproteins (*TNNC1*, *ABI3BP*), and proteoglycan (*OGN*). Using the patient demographic information, we further compared how patients designated into different risk group differ in terms of other clinical variables (Fig. 4c). The high-risk group consisted of patients with a slightly higher proportion of men, patients with squamous cell carcinoma, patients with smoking history, and stage II patients compared to low-risk group.

**The universal EPPI cutoff score for patient stratification.** To evaluate the clinical application of the EPPI risk score, we performed AUC analyses to determine a universal threshold that can be applied to all early-stage NSCLC patients. As different microarray platforms measure the same gene with varying expression levels on different scales and ranges[14], the most commonly available platform in a public repository of gene expression profiles was chosen to assess the potential of defining a universal EPPI risk score. The common cutoff value was determined as the median of the optimal cutoff scores at 5-year and 10-year survival, identified from the time-dependent AUC

analyses using the discovery set, as shown in Fig. 5a (cutoff score = 20.8). The cutoff was applied to stratify patients into low-risk and high-risk group in multiple independent data sets that were probed with the same platform (Fig. 5b). Figure 5c shows that the common threshold separated patients into groups with significantly different survival outcomes in both discovery and validation cohorts.

**Prediction of the adjuvant chemotherapy response.** We further tested the therapeutic predictive utility of the EPPI signature in an independent validation cohort, hypothesizing that the high-risk group would likely benefit from the ACT: 144 tumors from GSE42127 comprising 35 patients who received ACT and 109 patients who did not receive ACT. We systematically compared the predictive performance of our EPPI scoring system with known clinical prognostic markers using ROC/AUC analyses. The tested clinical variables included AJCC staging, histology, and gender. In order to assess the predictive accuracy of these models, time-dependent AUC analyses for 10-year survival were compared (Fig. 6a). The data suggest that the EPPI risk score has

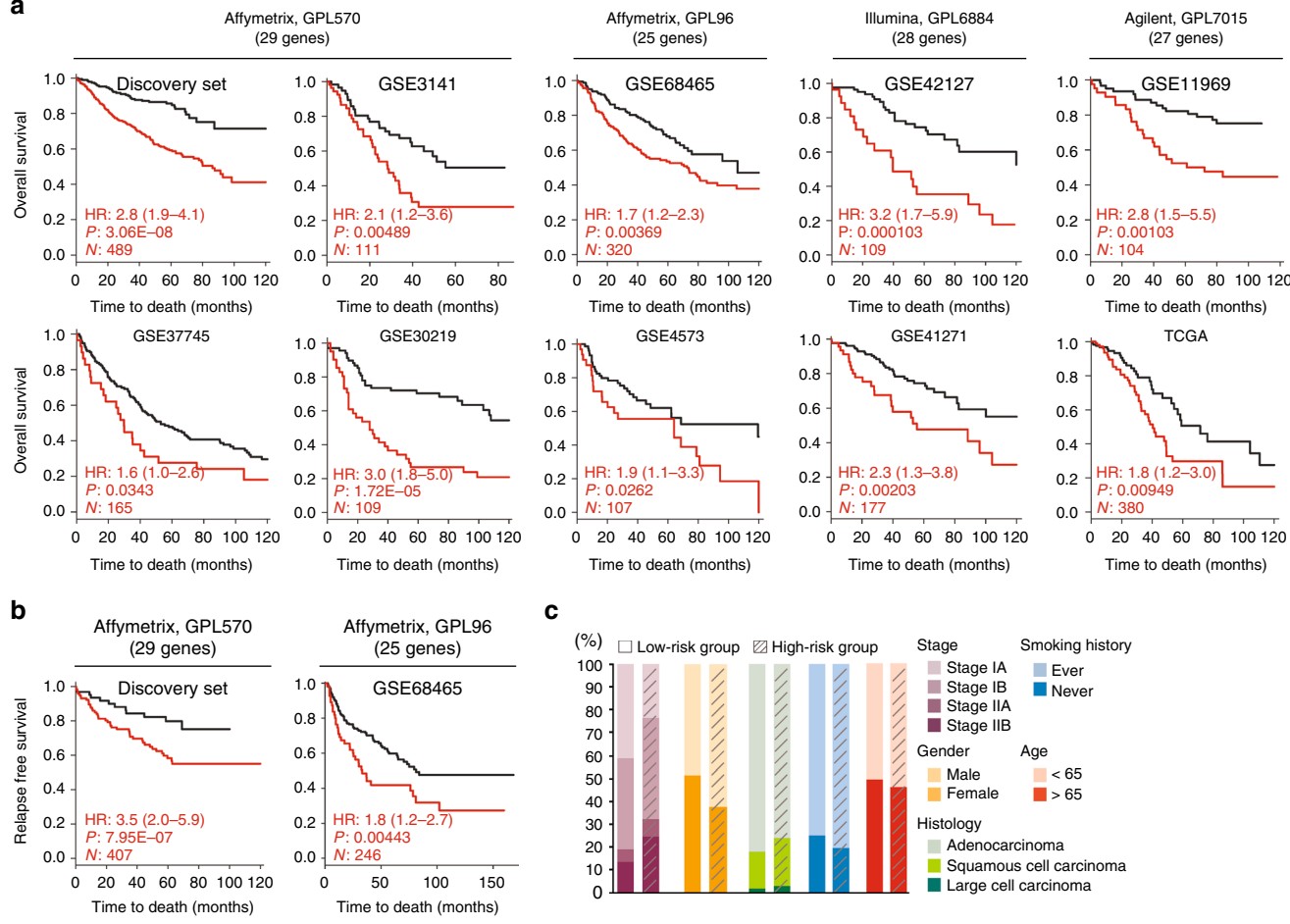

**Fig. 4** Prognostic value of the EPPI gene signature. The EPPI risk score stratified patients into two groups based on optimal cutoff score in each data set with significantly different **a** overall survival (OS); and **b** relapse free survival (RFS). The adjusted hazard ratio (HR), log-rank p-value (P), and the number of patients successfully stratified (N) determined from univariate Cox regression analyses are shown on each survival KM curve. Black and red KM curves represent predicted low-risk and high-risk group, respectively. **c** Bar graph comparing the number of patient for other clinical prognostic variables between high-and low-risk group

a best prognostic accuracy in predicting survival post surgery. With estimation of sensitivity and specificity using ROC curves, the optimal cutoff score was further determined to assign patients into different groups and identify those who may benefit from ACT (cutoff score = 21.7; Fig. 6b). Patients were ranked by their EPPI risk score and separated into low-risk and high-risk groups with the optimal cutoff score (Fig. 6c). Figure 6d shows that the OBS cohort experienced shorter survival than those with ACT in the high-risk group whereas patients who received ACT had no significant survival benefits relative to those without ACT in the low-risk group.

## Discussion

Although numerous gene panels have been claimed to be prognostic for early-stage NSCLC tumors (Supplementary Table 5), EPPI gene panel comprising specific ECM molecules has critical biological and clinical significance. As opposed to these existing signatures where genes were first ranked and selected solely based on their prognostic power using prior information of known survival outcomes, we were completely blinded to survival data when drawing EPPI signature; only existing biological knowledge was used for gene selection, particularly for biological motives representing ECM matrix that links tumor cells and surrounding enzymes in tumor microenvironment using multiple gene ontology databases. For this, we found it remarkable that early-

stage tumors of a particular kind of lung cancer, NSCLC, can be stratified with different prognosis based on ECM molecules alone, considering their highly variable and dynamic properties[15]. The diverse expression levels shown in each patient further suggests inter-tumor heterogeneity in relation to ECM components in the tumor microenvironment.

Potential involvement of these 29 specific ECM molecules in resistance to therapeutic intervention further highlights the significance of downstream analysis of ECM mediated signaling pathways in identifying druggable targets against NSCLC tumors. This finding is significant not just in demonstrating predictive value for prognosis but also in uncovering gene signature for the prioritization of tumor microenvironment-targeting therapeutic approaches. Despite immense efforts, designing rational therapy targeting cancer-associated fibroblasts (CAFs), for example, has been challenged and resulted in significant toxicity due to lack of definitive biomarkers in distinguishing 'activated' CAFs from normal fibroblasts[16]. Such lack of specificity and prior knowledge of tumor microenvironment at the molecular level truly necessitates specific ECM biomarkers to develop more precise and less toxic targeted therapies for medical utility. In this regard, our gene panel holds great promise in providing novel potential targets for stroma-directed therapeutic approaches—in fact, one of our ECM component, *COL11A1*, has very recently been reported as a highly specific biomarker of activated CAFs together

**Table 1 The EPPI signature is an independent predictor of survival outcome in early-stage NSCLC patients**

| Variables | GSE3141 HR (P) | GSE37745 HR (P) | GSE30219 HR (P) | GSE68465 HR (P) | GSE4573 HR (P) | GSE42127 HR (P) | GSE41271 HR (P) | GSE11969 HR (P) | TCGA HR (P) |
|---|---|---|---|---|---|---|---|---|---|
| *29-gene EPPI* | | | | | | | | | |
| Low-risk | 1 | 1 | 1 | 1 | 1 | 1 | 1 | 1 | 1 |
| High-risk | 2.4 (0.002) | 1.6 (0.09) | 2.9 (0.0006) | 1.9 (0.003) | 1.9 (0.04) | 3.0 (0.001) | 2.2 (0.006) | 2.5 (0.01) | 2.1 (0.005) |
| *Histology* | | | | | | | | | |
| ADC[a] | 1 | 1 | 1 | [b] | [c] | 1 | 1 | [d] | [d] |
| LCC[d] | | 0.96 (0.9) | 0 (1) | | | | 15.2 (0.01) | | |
| SCC[e] | 0.65 (0.12) | 0.93 (0.7) | 2.9 (0.001) | | | 0.99 (0.98) | 0.8 (0.6) | | |
| *Gender* | | | | | | | | | |
| Female | – | 1 | 1 | 1 | 1 | 1 | 1 | 1 | 1 |
| Male | | 0.93 (0.7) | 0.6 (1.2) | 1.1 (0.8) | 1.4 (0.3) | 0.98 (0.95) | 1.3 (0.4) | 1.2 (0.8) | 1.0 (0.96) |
| *Smoking* | | | | | | | | | |
| Never | – | – | – | 1 | 1 | – | 1 | 1 | 1 |
| Ever | | | | 0.9 (0.7) | 0.3 (0.1) | | 1.5 (0.5) | 0.9 (0.9) | 2.1 (0.15) |
| *AJCC Staging* | | | | | | | | | |
| IA | – | 1 | – | – | 1 | 1 | 1 | 1 | 1 |
| IB | | 1.5 (0.2) | | | 1.3 (0.4) | 1.2 (0.6) | 0.96 (0.9) | 1.6 (0.3) | 0.9 (0.8) |
| IIA | | 1.2 (0.8) | | | 2.3 (0.2) | 1.8 (0.3) | 1.1 (0.8) | 2.9 (0.2) | 2.6 (0.02) |
| IIB | | 1.6 (0.1) | | | 1.2 (0.7) | 1.2 (0.8) | 1.3 (0.6) | 2.10 (0.2) | 2.4 (0.02) |
| *Age* | – | 1.0 (0.02) | 1.02 (0.11) | 1.1 (1.4E-05) | 1.0 (0.7) | 1.0 (0.07) | 1.0 (0.03) | 1.0 (0.05) | 1.0 (0.02) |

–Data not provided
[a]Adenocarcinoma
[b]Exclusively comprising adenocarcinoma samples
[c]Exclusively comprising squamous cell carcinoma samples
[d]Large cell carcinoma
[e]Squamous cell carcinoma

with other co-expressed genes, which has further validated in 13 different types of cancers in TCGA data[16].

We have thoroughly revisited gene panels from 61 studies that have been previously claimed to be prognostically important (Supplementary Table 5). For fair comparison to our ECM gene panel, signatures selected based on biological motives were included for schematic demonstration (Supplementary Fig. 2). Here, only three factors that are considered to be most significant in developing clinically applicable multigene assay in a routine clinical setting are shown: (1). Hazard ratios for stratifying performance; (2). Validation sample size for statistical robustness; and (3). Number of genes used in the panel for feasibility and practicality. Some of these signatures with biological implications include cancer-related genes such as *TOP2A*[8,17,18], *PI3K*-related genes[17], and *VEGFA*[19], which are known to be significantly associated with clinopathological factors[20,21]. Even when comparing to these signatures, our *ECM* gene panel demonstrates comparable and even stronger prognostic performance in greater number of validation cohorts (Supplementary Fig. 2). Nevertheless, as Subramanian et al.[22] critically pointed out, hazard ratio is not sufficient enough to demonstrate predictive power; thus ROC curves were further obtained to demonstrate that EPPI risk score is a statistically better predictor of survival than known standard risk factors, including AJCC staging, histology, smoking status and gender, which were often missed in prior studies.

The differences observed in terms of traditional clinical factors between the low-risk and high-risk groups may exert undue bias, as shown from previously published studies[23]. In order to account for these clinical parameters, we performed multivariate Cox regression independently for each validation cohort adjusting for each clinical prognostic indicator to compute if these variables had any influence in assessing the predictive value of the EPPI. Consistently, the EPPI was shown to be the best prognostic factor among all the established clinical variables and remained to have statistical significance in predicting survival of early-stage NSCLC patients (Table 1). Direct comparison to traditional clinicopathological factors was further done and our EPPI gene panel

again remains to be better predictor of survival than known risk factors in most validation cohorts (Supplementary Fig. 3).

As our gene panel involves a minimal number of genes that can be quantified for its expression in a standard way using RT-PCR, genetic assays can be directly applied to FFPE specimens, which are usually available after surgical resection, for prospectively conducted validation study. In fact, Pervenio Lung RS and myPlan Lung Cancer have been tested with FFPE samples to derive risk score and demonstrated the feasibility of using FFPE tumor blocks or consecutive slides with defined criteria for tumor size and density[24]. In situations where samples have incomplete EPPI expression data, we have demonstrated that the clinical utility remains strong. Nevertheless, it is noteworthy that the use of the entire gene set is favored.

The developed genomic algorithm greatly reduced the computational complexity of the integrative genomic analysis. The stringent statistical criteria for differential expression analysis applied in this study narrowed down more than 20,000 genes to statistically significant and stable genes that may significantly contribute to cancer-associated molecular pathways, particularly in ECM remodeling. Interestingly, subsets of the genes in the EPPI had been previously reported for their differential expression in lung cancer or even other lung diseases such as idiopathic pulmonary fibrosis (IPF), chronic obstructive pulmonary disease (COPD), and cystic fibrosis (CF) when compared with normal lung tissues (Supplementary Table 4; Fig. 7a). The EPPI and previously identified differentially expressed genes from published transcriptome analysis of diseased lung tissues for IPF patients, in particular, demonstrate significant overlapping gene expression patterns. Among them, three genes, *SPP1*, *MMP1*, and *S100A2*, were strikingly upregulated and four genes, *FCN3*, *HHIP*, *S100A12*, and *CPB2*, were downregulated in both NSCLC and IPF. We thus speculate that the patterns of the EPPI gene expression might be a critical gauge for impaired lung function, and 29 genes in the EPPI might form a network and collectively mediate tumor initiation. The EPPI signature may further be associated with the underlying mechanism of carcinogenesis such

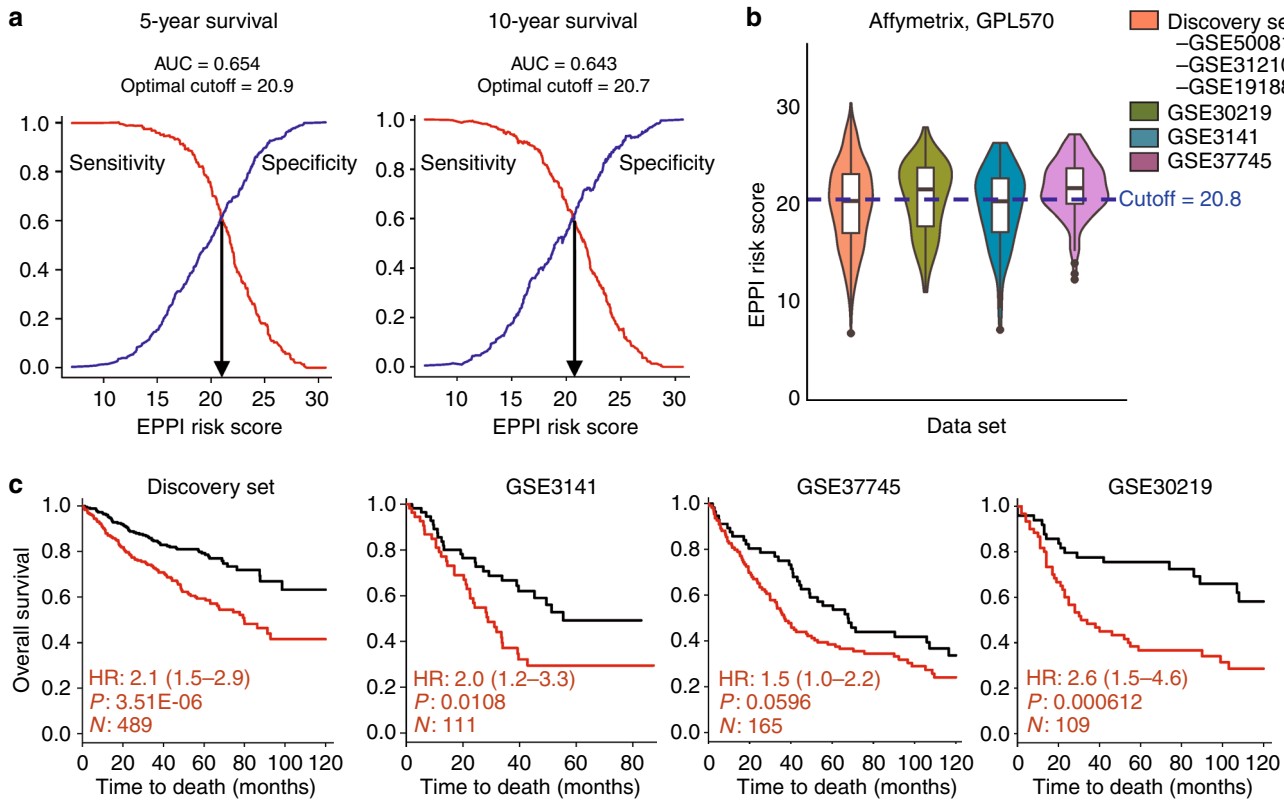

**Fig. 5** Identification and validation of a common cutoff score using AUC analyses. **a** Time-dependent survival ROC curves at 5-year and 10-year from the discovery set are shown with best trade-off between sensitivity and specificity. **b** Violin plot depicting the score distribution in each data set probed with Affymetrix-GPL570 platform; a cutoff score of 20.8 was used to stratify patients into 2 groups. **c** Validation of prognostic performance in multiple data sets probed in Affymetrix-GPL570 platforms. The adjusted hazard ratio (HR), p-value (P), and the number of patients successfully stratified (N) determined from univariate Cox regression analyses are shown on each survival KM curve. Black and red KM curves represent predicted low-risk and high-risk group, respectively

as epithelial-mesenchymal transition (EMT) or mesenchymal-epithelial transition (MET). For example, in vivo development of the alveolar epithelial EMT was found to be regulated by the ECM components during IPF[25]. It could therefore be valuable to investigate the association of each EPPI signature with the initiation of EMT, which may explain the fundamentals of tumor initiation.

Individual components of the EPPI have also been confirmed to have strong predictive value in other cancer types. Particularly, two genes of the EPPI gene signature, namely *SPP1* and *OGN*, which encodes ECM glycoprotein and proteoglycan, respectively, were identified among 26 stromal gene signatures that stratified disease outcome in primary breast tumors[12]. Consistent with these findings, 79 differentially expressed genes identified in another study[26] between primary breast cancers and paired lymph node metastases further overlapped with four genes from the EPPI: *COL11A1*, *CTHRC1*, *PCOLCE2*, and *OGN*. These differentially expressed genes were found to predict tumors with high risk of developing metastasis within 43 months, establishing their potential prognostic value in predicting clinical outcome of node-positive patients. Beyond the shared expression patterns of the EPPI gene signature in other lung diseases, the potential prognostic performance in other types of cancer further provide the possible association of cancer progression to be related to ECM-dependent molecular pathways.

Prior assessment of primary breast carcinomas and matching lymph node metastases[27] revealed ECM molecules, comprising 18% of differentially expressed genes, to be most differentially enriched. Additionally, a recent single-cell RNA-seq study[28]

found unexpected high abundance of ECM transcripts in substantial proportion of enriched circulating tumor cells (CTCs), which were shed from matched primary pancreatic tumors into the bloodstream. The knockdown of one of highly expressed ECM proteins in cancer cells, *SPARC*, suppressed cell migration and invasion, suggesting the potential role of CTCs in the classical "seed" carrying their own "soil" to metastasize[28]. Most interestingly, four genes of our EPPI overlapped with their differentially expressed genes in CTCs vs. matched primary pancreatic tumors: *SPP1*, *ABI3BP*, *OGN*, and *SFTPD*. While this study has demonstrated potential benefits of applying the EPPI as a selection tool for ACT in patients at high risk of relapse, this further remains to be validated in larger subsets of patients or in a prospectively conducted study.

One of the limitations of this study is the small sample size used to predict the survival benefits of ACT using the EPPI risk score. This is due to the incomplete demographic information in the many available public data sets. We therefore plan to assess the EPPI signature in a larger validation data set cohort. This will aid to establish the cutoff value over a wider range of patient subtypes. Another limitation is the cross platform compatibility of our EPPI risk scoring matrix, as other platforms use different scale range and have varying gene expression values. Nevertheless, we established our work using the most widely adopted platforms, demonstrating its successful classifications of early-stage patients. In our attempt to meet the criteria set by Subramanian et al.[22] for a clinically applicable gene expression-based prognostic signature, we have thoroughly revisited and analyzed our findings again and believe that the EPPI gene panel

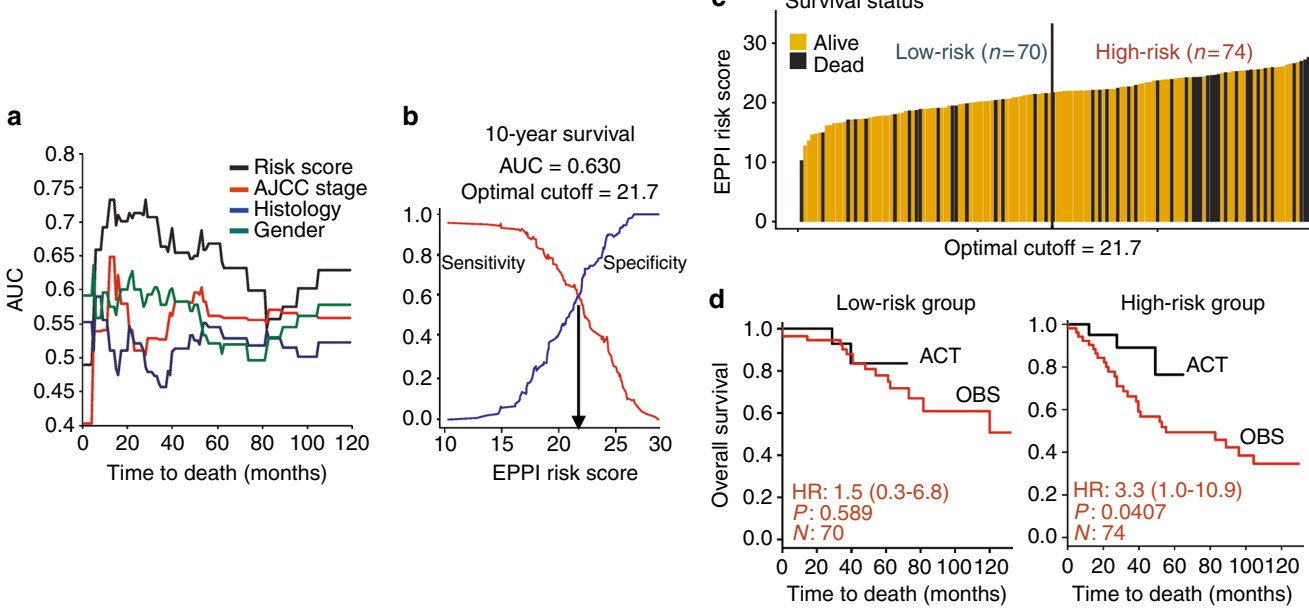

**Fig. 6** Predictive value of the EPPI gene signature. **a** Time-dependent AUC analysis of EPPI risk score and other clinical variables in GSE42127 regardless of adjuvant chemotherapy (ACT) history. **b** Time-dependent ROC curves at 10 year are shown with best trade-off between sensitivity and specificity (optimal cutoff = 21.7). **c** The EPPI risk score distribution, survival status and patient stratification based on optimal cutoff determined from ROC analysis are shown (high-risk vs. low-risk group). **d** Hazard ratio (HR), p-value (P), and the number of patients successfully classified (N) determined from univariate Cox regression survival analyses are shown. Black and red KM curves represent ACT and OBS (observation group: patients without ACT) group, respectively

fulfill most of the stated standards (Fig. 7b). This shows the promising results for its clinical application and our informatics analyses can be easily translated to other platforms as well.

Although our current universal cutoff score was determined at the intersection point of sensitivity and specificity, this score may not always be the most optimal point in the clinical practice; we believe that achieving higher sensitivity is more critical (even at the expense of having low specificity; that is producing a higher proportion of false-positive results) as treatment and survival outcomes greatly depends on the stage of cancer at diagnosis in NSCLC. If patients are put at considerable risk by receiving false-positive results, it may be reasonable to compromise sensitivity—in which we believe is less significant than identifying more patients with early-stage cancer for greater benefits. Taken together, our study demonstrated a critical and important analysis technique that will complement current disease management. With potential role of ECM proteins in cancer progression highlighted, we believe understanding the role of specific EPPI gene molecules in metastasis may further provide potential targets for future intervention and therapy. A prospective randomized clinical interventional study based on the EPPI signature will better confirm our analysis to maximize the benefit of the treatment administered to patients.

## Methods

**Microarray data.** Transcriptional profiles and clinical annotations were downloaded from the National Center for Biotechnology Information Gene Expression Omnibus (GEO, http://www.ncbi.nlm.nih.gov/geo) via the inSilicoDb package[29] in R/Bioconductor. In order to minimize undesired variations from different platforms, only samples probed with Affymetrix Human Genome U133 Plus 2.0 (HG-U133_Plus_2) Array were selected as the discovery set. Manual curation was done to include only primary NSCLC tumors such as adenocarcinoma, squamous cell carcinoma (SCC), and large cell carcinoma (LCC) and normal lung tissues. Ten primary NSCLC tumor data sets (Fig. 2a) from the GEO repository were downloaded, pre-processed and normalized using frozen Robust Multi-array Average (fRMA).

Since combining multiple data sets into one large-scale analysis carry over non-biological experimental variations or batch-effects[30], direct adjustment for the undesired batch effects with the Empirical Bayes algorithm implemented in

ComBat was performed in these fRMA-normalized NSCLC data sets using the inSilicoMerging package[31]. Principal Component Analysis (PCA) was carried out using the prcomp function to collapse high-dimensional data into the first two components and they were visualized using via the ggbiplot package[32] in order to validate the batch-effect removal in Combat-transformed data. A total cohort of 1118 human samples including 925 primary NSCLC tumors and 193 normal lung epithelial tissues were generated for the identification of differentially expressed genes. The genefilter package[33] was subsequently used to apply a filter to remove genes with low variance across samples, improving the computational processing time by focusing only on statistically significant genes.

**Construction of the EPPI signature.** Differential expression was assessed by a linear regression method using the R/Bioconductor *limma* package[34]. We applied stringent statistical cutoffs of log2FC > 1.5 and adjusted p-value < 1.0E-10 for genes to be determined as differentially expressed in the integrated primary NSCLC tumor data for subsequent discovery and analysis. Volcano plot was generated using the ggplot2 package[35] to graphically reveal genes that differ significantly between two phenotypes of diseased and normal samples. Limma-generated differentially expressed genes that met our statistical criteria were filtered to construct the ECM-related gene list using the Matrisome database[36], Gene Ontology (GO) Consortium[13] and Kyoto Encyclopedia of Genes and Genomes (KEGG) pathway database[37]. A descriptive list of 29 EPPI genes including the fold change (log2-base) and adjusted p-value (false discovery rate) is shown in Supplementary Table 2.

**RNA-Seq data.** Level-3 RNAseqV2 gene expression profiles were downloaded from the Cancer Genome Atlas (TCGA) using the TCGA-Assembler package[38]. The data comprised clinical information and gene expression values of 1018 primary tumors and 110 normal lung tissues from patients with adenocarcinoma (LUAD) and squamous cell carcinoma (LUSC). The raw data were processed with RNA-Seq by expectancy maximization (RSEM), and genes without least zero RSEM count in at least 20% of the samples were removed for further differential expression analysis. Differential expression analysis was performed using the voom function in the limma package, with normalization by Trimmed Mean of M-values (TMM) via the edgeR package[39].

**Gene set enrichment analysis.** The enrichment of identified upregulated and downregulated gene set identified using the discovery set was assessed in TCGA cohorts using GSEA v2.2.2. GSEA computes the enrichment score by applying weighted Kolmogorov–Smirnov statistic to a running sum of the ranked list with 1000 permutations. The enrichment score (ES) was further normalized to account for the size of each inputted gene set. The false discovery rates (FDR) <0.001 were assumed to be statistically significant.

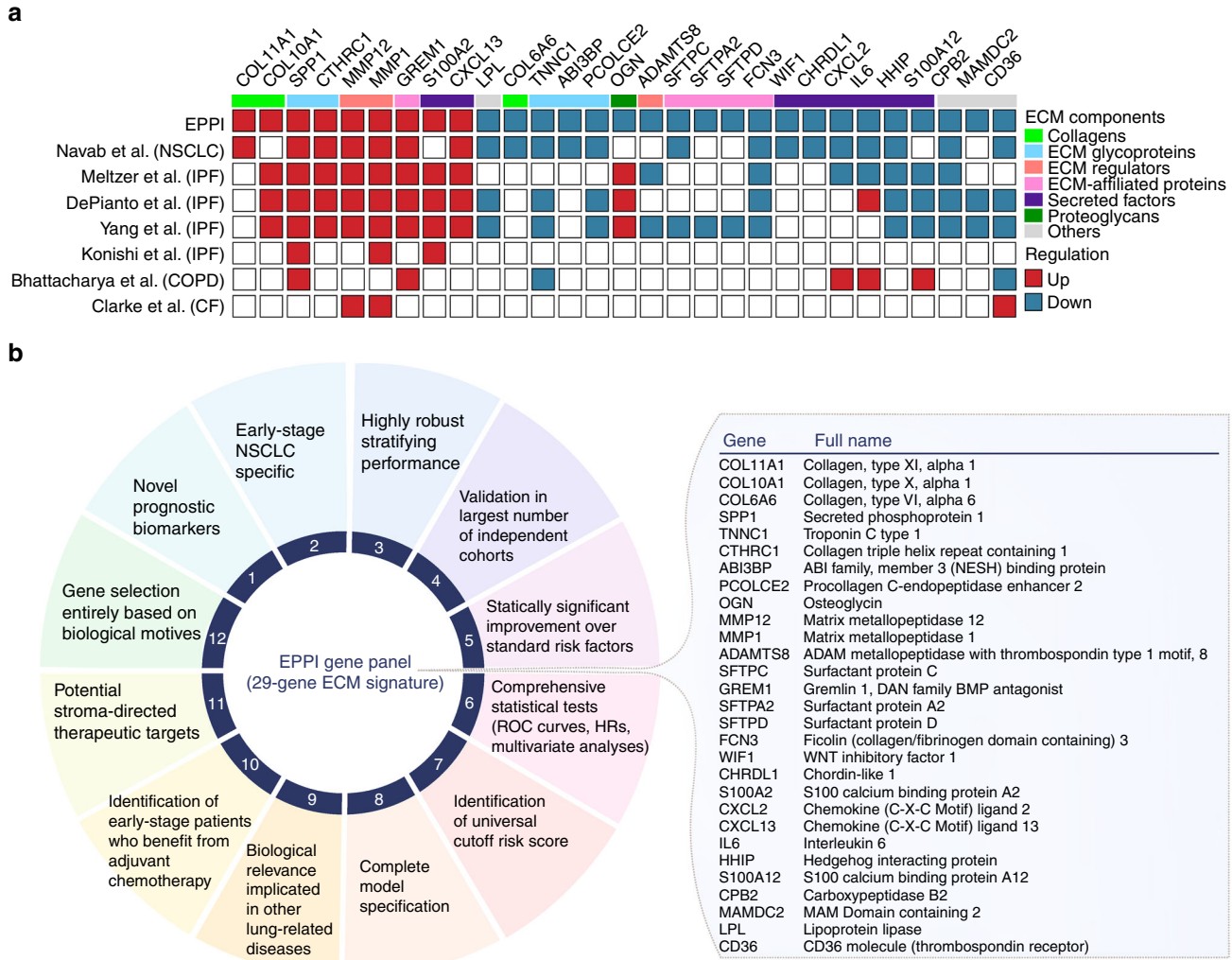

**Fig. 7** Biological and clinical significance of the EPPI gene signature. **a** Schematic representation of the expression patterns of the EPPI gene signature in lung cancer and other lung-related diseases. Differential expression levels in prior works were calculated using their own statistical methods from diseased lung tissues vs. matched normal lung tissues. **b** Summary of our gene panel in terms of critical factors to be considered in developing gene expression-based prognostic signatures

**EPPI classifier validation and its prognostic evaluation**. To validate the prognostic value of the EPPI in predicting OS and RFS, normalized gene expression profiles (log2 transformed) from eight independent microarray data sets across four different platforms of Affymetrix (HG-U133A and HG-U133_Plus_2), Illumina (HumanWG-6 v3.0 Expression), and Agilent (014850 Whole Human Genome Microarray 4 × 44K G4112F) were downloaded via the GEO repository directly. For all validation cohorts, manual curation was done to select only primary tumors resected from stage IA to IIB (early-stage) patients and the survival data were censored at 10 years follow-up post surgery. Using a Spearman rank correlation as a distance metric selection, an unsupervised hierarchical clustering algorithm was first performed to test the prognostic value of the EPPI signature in the validation cohort (Supplementary Fig. 1) using the Institute of Genomic Research MultiExperiment Viewer (TIGR MeV) version 10.2 software. The EPPI risk score of each patient was then computed and used to divide patients into low and high-risk group based on the optimal cutoff score identified in each data set. The optimal cutoff EPPI risk score, which is defined as the score with the most significant split using log-rank test, was determined using the Cutoff Finder algorithm[40].

**Personalized EPPI risk score and gene expression heat maps**. The regression coefficient of each EPPI gene was derived from 489 early-stage NSCLC patients who had complete survival information from the initial discovery set. The EPPI risk score was calculated as a sum of multiplication of the expression level of each EPPI gene and its respective Cox regression coefficient identified from the discovery set (EPPI risk score = 29+∑ (expression level of gene) X (Cox regression coefficient)). Each Cox regression coefficient used for EPPI risk score computation is shown in Supplementary Table 3. A constant was added to ensure all scores are positive. The risk score generated for each patient in the discovery set was visualized with a personalized two-dimensional heatmap using Gene-E open software (Broad Institute).

**Time-dependent ROC, AUC, and survival analyses**. Time-dependent receiver operator characteristic (ROC) and area under the curve (AUC) were computed using the survivalROC package[41] in R. OS and RFS time were computed from the date of surgery until death and relapse of last follow-up contact, respectively. For all survival analyses in this study, both univariate and multivariate Cox proportional hazards model and Kaplan–Meier (KM) survival curves were used via the survival package in R/Bioconductor[42]. Hazard ratios were adjusted for available clinical variables in each data set such as smoking status, gender, age at diagnosis, histology and tumor stage. The Cox proportional hazard assumption was checked by Schoenfeld test using cox.zph function in survival library[42].

**Data availability**. Data for all the bioinformatics analyses using open-source R packages have been described throughout the text and are available under accession codes GSE10245, GSE10445, GSE10799, GSE12667, GSE18842, GSE19188, GSE28571, GSE31210, GSE33356, GSE50081, GSE3141, GSE37745, GSE30219, GSE68465, GSE4573, GSE42127, GSE41271, and GSE11969 from the National Center for Biotechnology Information Gene Expression Omnibus (GEO). All statistical analyses have been performed using open-source R packages and described in Methods section. All other data associated with this work are available in the Supplementary Information or from the corresponding author upon reasonable request.

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

## Acknowledgements

This work was conceived and carried out at the MechanoBioEngineering laboratory at the Department of Biomedical Engineering, National University of Singapore (NUS). We thank both present and former members of the MechanoBioEngineering lab for their support and assistance. S.B.L. acknowledges scholarship and support from NUS Graduate School for Integrative Sciences and Engineering (NGS). Support provided by the National Research Foundation, Prime Minister's Office, Singapore under its Research Centre of Excellence, Mechanobiology Institute at NUS is also acknowledged.

## Author contributions

S.B.L., S.J.T. and C.T.L. designed the study. S.B.L. performed bioinformatics analyses. S.B.L., S.J.T., W.T.L. and C.T.L. analyzed and interpreted the data. S.B.L., S.J.T., W.T.L. and C.T.L. reviewed and contributed to the manuscript.

## Additional information

**Competing interests:** The authors declare no competing financial interests.

