## [Peer Review File · Nature Communications]

Reviewers' comments:

Reviewer #1 (Remarks to the Author): Expert in NSCLC

A gene expression signature based on 29 ECM related genes (EPPI) was established by comparing NSCLC gene expression data and data sets from normal lung tissue. This gene expression signature was able to predict survival in 10 independent data sets and was suggested to be of predictive value for the benefit of adjuvant chemotherapy.

This study is one of numerous studies claiming to identify a prognostic signature that is better than other traditional stratification and is completely based on in-silico analysis of published studies. Also the focus on stromal markers is not new (Navab et al., 2011, PNAS; Edlund et al., 2012, Int J Cancer) neither for other cancer forms nor for lung cancer. The study does not prove that the EPPI is better than a combination of important clinical parameters and does also not suggest how this signature can be implemented in diagnostic.

The main bioinformatic approach of this study warrants a careful evaluation by a biostatistician.

There are several more aspects that should be addressed:

The selection of gene expression data sets is not clear. There are several more data sets that can be used (e.g. Shedden et al.; GSE19188; GSE30219; GSE3141; GSE37745). What was the result of the curation of the data sets? How many cases were excluded?

I understood that the 29-gene signature was selected by the comparison of normal versus cancer tissue. How many genes (in absolute numbers) overlap between the microarray analysis and the RNAseq data?

On what samples/data sets were the prognostic model developed? I could not find a model description and also on which cohorts (training cohort?). How was the cut-off established (high risk versus low risk) that was applied in 10 independent validation cohorts? If the prognostic model is established on the tissue data sets (Table 5) the validation data set is not independent!

The validation data sets included only patients without adjuvant treatment (ACT). Why were these cohorts not used later to evaluate the predictive power for ACT, but only two very small cohorts?

The risk score should be directly compared to a combination of relevant traditional markers: the strongest are stage and performance status, the latter was not included in one of the multivariate analyses and is even not discussed. Also stage was only available in some of the cohorts. Notable in the high -risk score group were more patient with higher stage, older patient, more smokers, more female and more often adenocarcinoma patients, thus all parameters associated with poor prognosis. Although the risk score seems to be better in the multivariate Cox regression models of each single study a combination of these clinical parameters was not compared.

The selection of both adjuvant studies GSE42127 and GSE47115 is not described sufficiently. These are retrospective cohorts not based on randomized trials, thus the selection of adjuvant chemotherapy is primarily biased and interpretation is difficult. In addition, the risk score did not clearly show the value of the ACT of high risk score patients ($p=0.16$, lower figure)?

The ROC curves again showed the prognostic performance of the both data sets, already indicated in figure 3 and table 1. It does not add information on the adjuvant aspect.

The discussion do not address important aspects of the field.

What is the advantage of this risk score to other prognostic signatures?

What is the overlap of genes included in this risk score to other signauteres?

What is the practical effect of the application of the risk score? Is it possible to spare patients from adjuvant therapy? Can this method be applied in clinical practice? Is it feasible to perform gene expression arrays for each operated patient on fresh frozen tissue?

Reviewer #2 (Remarks to the Author): Expert in cancer gene expression and prognosis

A. The paper presents an ECM-related prognostic and predictive indicator (EPPI), based on the expression values of 29 genes in NSCLC patients. The EPPI is generated by calculating the tumor vs normal fold change (FC) for a microarray dataset of 925 tumor and 193 normal samples (n=1118), assembled from several independent datasets. 103 differentially expressed genes (DEGs) were identified. Next, gene set enrichment analysis was performed on these 103 genes (separately on up and down regulated genes) on RNAseq data from TCGA datasets (n=1128); a group of 29 ECM-related genes constituted the largest significantly enriched "cellular component". The EPPI risk score was defined as "the geometric mean ratio of the expression values of highly to lowly expressed gene clusters". The prognostic and predictive performance of the risk score was tested on a large number of independent cohorts, using the median value of the EPPI to separate the tumor samples of each cohort into high and low risk groups. Good performance was found.

B. The literature abounds with prognostic and predictive gene lists and associated risk scores for various cancers. This paper has in my opinion three significant advantages. 1. Existing biological knowledge is used for gene selection, as opposed to blindly ranking genes on the basis of some figure of merit. 2. The number of samples is large and many independent datasets are tested, indicating robustness of the success rate of the risk score. 3. The main focus is on a particular kind of lung cancer - NSCLC.

C. The data and analysis presented in the paper are solid and of high quality.

D. The statistical analysis is professionally done.

I do have a few points of criticism and suggestions that the authors should relate to and perform.

E. Robustness of prognostic gene lists has been a central issue in evaluation of studies of this type, see Ein Dor et al *Bioinformatics* vol 21 p 171 (2005). The overlap between "prognostic gene sets" produced by different studies tends to be dismally low. The present study uses previous knowledge (Gene Ontologies, databases to assign genes to biological function) and hence it has a much better chance of producing a robust gene set. I strongly recommend testing this hypothesis: use the two independent datasets n=1118 and n=1128) in a similar fashion (identifying DEGs, perform enrichment analysis) to produce two most enriched gene sets: see whether one gets the same genes when this is done, or - report the overlap between the two gene lists found. This is an essential step to dispel the existing suspicions against feature selection methods based on ranking genes.

F1. I find the definition of the EPPI risk score ambiguous - I think I can guess what they are doing, but I would feel much more comfortable with a more careful definition and even a formula.

F2. As I understand the manner in which the score is tested and applied, one needs an entire cohort to assign a sample to high/low risk groups (by the median over the cohort). This makes clinical application of the score rather difficult. How would setting a "universal" threshold, on the basis of the "training" phase, and using it in all tests, affect the quality of the prognostic and predictive results?

Reviewer #3 (Remarks to the Author): Statistical expert

Early-stage lung cancer is a disease setting where a biomarker for that would identify a subset who would benefit from adjuvant chemotherapy has been of interest for some time. Currently, patients generally don't get adjuvant chemotherapy. Several studies have attempted to do this and this current study is also addressing this problem. The authors claim to have had some success and if this is true that would be a potentially important finding. In particular, the authors

focused on a extracellular matrix associated signature consisting of 29 genes.

The interest in this study is that the authors have gathered together a large number of samples' data from many different sources. They also provide some justification for the focus on ECM-associated genes.

In general, the data and methodology used are strong, although there are some aspects of the study which dampen enthusiasm for the findings. One is that, in each dataset, different cutpoints were used for the classification (based on the median signature score). The manuscript would have been greatly strengthened if some attempt was made to evaluate the potential of a common cutpoint decision rule for all patients.

A second issue is the treatment of tumor stage. It was not clear how the staging was done. First, the version of the AJCC scoring system that was used for tumor staging needs to be given, since the system evolves over time. Also, there was not a distinction made between stage IA and IB, which are prognostically fairly different, and this data is generally available and therefore relevant for the multivariate analysis which purports to tell us whether provided information is novel.

A third issue is that the patients who received ACT were not randomized to it, presumably, which complicates the interpretation of the Kaplan Meier curves in Figure 4. (Also, relatedly, why is not an interaction p-value provided here?)

The authors also state on line 194 that the implementation of the EPPI signature in the clinical setting is straightforward. This suggests a lack of input from clinicians on the manuscript writing at least. More specifically, some discussion of the potential patient population and clinical setting this project is aiming towards seems needed.

In Table 1, why are there areas with missing data. This needs to be explained. Was there a model selection step, or is the data missing in the original datasets? More generally, the level of missing covariate data in each dataset should be provided somewhere.

The authors also state on line 164 that the "...best trade-off between sensitivity and specificity was ...". But the best tradeoff depends on clinical factors like prevalence, cost, side-effects, etc. Clinical context is needed for such claims.

Response to Comments from Reviewer #1 (Expert in NSCLC)

Comment: A gene expression signature based on 29 ECM related genes (EPPI) was established by comparing NSCLC gene expression data and data sets from normal lung tissue. This gene expression signature was able to predict survival in 10 independent data sets and was suggested to be of predictive value for the benefit of adjuvant chemotherapy.

This study is one of numerous studies claiming to identify a prognostic signature that is better than other traditional stratification and is completely based on in-silico analysis of published studies. Also the focus on stromal markers is not new (Navab et al., 2011, PNAS; Edlund et al., 2012, Int J Cancer) neither for other cancer forms nor for lung cancer. The study does not prove that the EPPI is better than a combination of important clinical parameters and does also not suggest how this signature can be implemented in diagnostic.

Response: We greatly appreciate all your comments and suggestions that

allowed us to thoroughly analyze our study again and made significant changes that added further value to our current scientific finding. We agree that the study is one of many studies claiming prognostic value of specific genes in cancer patients.

Novel findings in this study, however, are fundamentally different from that in prior studies:

1. Prior genetic markers are generally focused on oncogenes.
2. Our work highlighted the significance of a unique set of genes exclusively comprising of ECM signatures.
3. ECM is closely related to the tumor microenvironment.
4. ECM components differ from carcinoma-associated fibroblasts (CAFs) genes (Navab et al, 2011) and CD99 (Edlund et al., 2012).

- ❖ 11 prognostic CAF genes Navab reported is only a subset of stromal components, and there is no overlapping gene with our EPPI signature (11 CAF genes: *ICAM-1*, *THBS2*, *MME*, *OXTR*, *PDE3B*, *CLU*, *B3GALT2*, *EVI2B*, *COL14A1*, *GAL*, and *MCTP2*) where 29-gene EPPI mostly consists of surfactants, MMPs, collagens, ECM glycoproteins, proteoglycans and secreted factors. Taken together, our study highlighted the significance of new prognostic ECM gene signatures that may provide potential targets for future interventions and therapy.

- ❖ Second study reviewer#1 mentioned (Edlund et al., 2012) reported prognostic value of *CD99* in NSCLC at protein level. The prognostic performance of protein expression of one specific molecule does not equate to collective prognostic value of ECM components at mRNA-level. Moreover, our EPPI signature does not include *CD99*.

5. Target patients are exclusively comprised of early-stage NSCLC.

- ❖ The validation patient pool of prior studies claiming prognostic signatures include patients from Stage I to Stage IV. More distinct prognostic value (smaller *P* value in survival analysis) is usually observed when including both early-stage patients (Stage I & II) and

late-stage (Stage III&IV) as high-risk groups defined with gene signature will mostly comprise of late-stage patients. To facilitate the clinical utility of EPPI signature, our validation cohort only consisted of early-stage patients (Stage I&II). This may greatly assist physicians in identifying early-stage patients who can be spared the toxicity and side effects of chemotherapy after tumor resection.

6. Gene set that we defined is unique and further adds on to show the importance of the tumor microenvironment.

Unlike commercially available genomic kit for early-stage breast cancer patients (Prosigna, Mammaprint, Oncotype DX), there has been a lack of multi-gene diagnostic assays for early-stage NSCLC patients due to its heterogeneity, poorly overlapping genes which are claimed to be prognostic, and lack of validation in large patient pool. Such limitations truly motivated us to use 'big data' that are available in public genomic repositories to establish largest validation cohort using robust bioinformatics workflow.

Comment: The main bioinformatics approach of this study warrants a careful evaluation by a biostatistician. There are several more aspects that should be addressed: The selection of gene expression data sets is not clear. There are several more data sets that can be used (e.g. Shedden et al.; GSE19188; GSE30219; GSE3141; GSE37745). What was the result of the curation of the data sets?

Response: We truly appreciate your careful examination and inclusion of other available references. In response to this, we have included all the datasets you have mentioned in our validation cohort. We first defined "discovery cohort" and "validation cohort" for differential expression analysis and prognostic performance validation respectively.

The initial discovery cohort (10 datasets) in identifying DEGs consists of

GSE10245, GSE10445, GSE10799, GSE12667, GSE18842, GSE19188, GSE28571, GSE31210, GSE33356, and GSE50081 which were all probed with same platform, GPL570 [HG-U133_PLUS 2] Affymetrix Human Genome U133 Plus 2.0 Array. Selection of datasets with same platform allow us to perform Combat correction for integration and thus to minimize gene loss during statistical correction.

Validation cohort consists of discovery cohort (datasets with survival data among 10 datasets: GSE50081, GSE31210, GSE19188), and independent datasets (datasets not included in discovery cohort: GSE3141, GSE37745, GSE30219, GSE68465, GSE4573, GSE42127, GSE41271, GSE11969, and TCGA).

As the reviewer suggested, GSE19188, GSE30219, GSE3141 and GSE37745 were all included in our validation cohort. As demonstrated by consistently high hazard ratios across all validation cohorts, there was a significant association between survival and EPPI gene panel in early-stage NSCLC patients.

Comment: How many cases were excluded?

Response: For all datasets in validation cohort, only early-stage (Stage I&II) and NSCLC patients with histology Adenocarcinoma (ADE), Squamous Cell Carcinoma (SCC) and Large Cell Carcinoma (LCC) were included. Patients without survival data such as survival status or survival time period were excluded (e.g. overall survival = n.a.).

Validation cohort:

1. Discovery set (GSE50081, GSE31210 and GSE19188 were the only datasets among 10 datasets that contain survival data).

GSE50081: 0 excluded (n= 181)

GSE31210: 0 excluded (n= 226)

GSE19188: 0 excluded (n=82)

Thus, integrated (fRMA normalized-Combat adjusted) discovery set consists of a total of 489 patients in validation cohort.

2. GSE3141: 0 excluded (n=111)

3. GSE37745: 31 excluded (n= 197 – 31 = 165)

Only patients with Stage I&II were included, and the rest (Stage III&IV) were excluded.

4. GSE30219: 184 excluded (n = 293 - 184 = 109)

Only patients with stated histology = ADE, SCC and LCC were included, and the rest with other histology such as SQC, LCNE, CARCI, BAS, and others were excluded.

5. GSE68465: 142 excluded (n = 462 - 142 = 320)

Only lung adenocarcinomas (normal lung tissue samples were excluded) with overall survival data (patients with OS=n.a. were excluded) and ACT/ART='no' were included.

6. GSE4573: 23 excluded (n = 130 - 23= 107)

Only patients with Stage I&II were included.

7. GSE42127: 67 excluded (n = 176 - 67 = 109)

Only patients with Stage I&II and ACT = 'no' were included.

8. GSE41271: 94 excluded (n = 271 - 94 = 177)

Only patients with Stage I&II and histology = ADE, SCC and LCC were included.

9. GSE11969: 59 excluded (n = 163 - 59 = 104)

Only patients with Stage I&II were included.

10. TCGA: 8 excluded (n = 388 - 8 = 380)

Only patients with Stage Ia, Stage Ib, Stage IIa, and Stage IIb and lung adenocarcinoma histology were included. In order to perform multivariate analysis and to include staging as one of clinical parameters (stage Ia, Ib, IIa, IIb), patients annotated with Stage I, Stage II without a or b and NA were excluded as follow: TCGA.49.AAQV: stage II, TCGA.49.AARQ:stage I, TCGA.50.6673: stage I, TCGA.67.4679: NA, TCGA.69.7980: stage I, TCGA.69.8254:NA, TCGA.97.8552: stage I, TCGA.L4.A4E5: stage I).

First 10 independent datasets (initial discovery cohort) were randomly selected regardless of having survival data (survival status and time period) to perform differential expression analysis. We further performed random sampling (10,000 iterations) using these 10 datasets to compute the overlap between DEG list from each iteration and thus to make certain that our final DEG list is not a result of any kind of biased selection. Moreover, our final DEGs were further validated in RNAseq data using Gene Set Enrichment Analysis (Fig. 2g).

Comment: I understood that the 29-gene signature was selected by the comparison of normal versus cancer tissue. How many genes (in absolute numbers) overlap between the microarray analysis and the RNAseq data?

Response: Applying the same statistical criteria to define DEGs using RNA-seq data, the proportion of DEGs is much greater than that of DEGs defined using microarray data, as can be seen in Fig. 2f. As for RNA-seq data genes without least zero RSEM count in at least 20% of the samples were removed during quality control check, thus not all DEGs defined using microarray data are present in the final DEG list from RNA-seq data.

Among our final DEG list (microarray) consisting of 103 genes, 23 genes were filtered during QC check and thus not found in RNA-seq data (*GKN2*, *GJB2*, *MAMDC2*, *HS6ST2*, *MCEMP1*, *PEBP4*, *INMT*, *ANLN*, *SCGB3A2*, *LRRK2*, *CTHRC1*, *HHIP*, *COL6A6*, *C2orf40*, *VEPH1*, *FAM150B*, *CDCA7*, *ADIRF*, *BTNL9*, *SOX7*, *SUSD2*, *ANKRD29*, and *NCKAP5*). Rest of 80 DEGs were all found to be differentially expressed using RNA-seq platform, further validating the robustness of differential expression of our final DEG list in NSCLC tumor samples.

Comment: On what samples/data sets were the prognostic model developed? I could not find a model description and also on which cohorts (training cohort?).

Response: We greatly appreciate your suggestion on defining training (stated as

'discovery cohort' in this study) and independent validation cohort. We have made significant changes in developing EPPI risk scoring metrics as follow:

1. The 'discovery cohort' was first established from initial 10 datasets where only early-stage NSCLC tumor samples with complete survival information were selected (from initial 925 primary NSCLC tumors from 10 datasets, 489 samples met these criteria).

2. The 'discovery cohort' was then used to compute the Cox regression coefficient for each EPPI gene (Supplementary Table 3, Fig. 3b). The EPPI risk score was then calculated using these computed coefficients with the gene expression values of 29 EPPI genes :

$$\text{Formula} = 29 + \sum (\text{Regression Coefficient}) * (\text{Gene Expression})$$

Using the defined regression coefficient for each EPPI gene, we could generate personalized EPPI gene expression heat-map using any validation dataset. An example using the discovery set is shown in Fig. 3a with the computed EPPI score distribution illustrated below. An optimal cut-off EPPI risk score that stratifies patients into low- and high-risk groups was further statistically evaluated with thorough ROC/AUC analyses.

Comment: How was the cut-off established (high risk versus low risk) that was applied in 10 independent validation cohorts?

Response: In response to this, we have made significant improvements on the algorithms for optimal cut-off determination in each dataset. The optimal cut-off score was defined as the score with the most significant split using log-rank test. It was computed using the Cutoff Finder algorithm [1].

Comment: If the prognostic model is established on the tissue data sets (Table 5) the validation data set is not independent!

Response: The datasets we used to develop the prognostic model were GSE10245, GSE10445, GSE10799, GSE12667, GSE18842, GSE19188, GSE28571, GSE31210, GSE33356, and GSE50081 (Fig. 1a). All datasets in our validation cohort are independent: GSE3141, GSE37745, GSE30219, GSE68465, GSE4573, GSE42127, GSE41271, GSE11969, and TCGA (Fig. 4a).

Comment: The validation data sets included only patients without adjuvant treatment (ACT). Why were these cohorts not used later to evaluate the predictive power for ACT, but only two very small cohorts?

Response: We understand that the limitation of our study is limited dataset available in public repositories to evaluate the predictive power for ACT. Among all datasets we used to develop the prognostic model in this study including both discovery and validation datasets, only GSE68465 and GSE42127 had information on the ACT history and thus we were able to filter patients who met our defined criteria for analyses. GSE68465, however, had information on adjuvant radiotherapy (ART) history as well. The presence of ART history may bias the results for the evaluation of the predictive power. Moreover, if we only consider patients who received ACT and not ART, only 43 patients could be selected for analysis in evaluating the predictive power for ACT. In this regard, we decided only to use GSE42127 to show the potential predictive power of the EPPI for ACT in 144 patients altogether.

Comment: The risk score should be directly compared to a combination of relevant traditional markers: the strongest are stage and performance status, the latter was not included in one of the multivariate analyses and is even not discussed. Also stage was only available in some of the cohorts.

Response: First of all, we greatly appreciate your opinions on this matter. We have thoroughly performed the multivariate analyses in all independent validation cohorts to adjust for any bias that other traditional clinical parameters may bring

in evaluating prognostic performance of the EPPI. We included all the clinical information that the dataset provided in our multivariate results to make completely certain that the EPPI had robust prognostic performance.

As for the staging variable, fairly large proportion of validation datasets with stage-adjusted results was included in our multivariate analysis (6 out of 9 independent validation datasets: GSE37745, GSE4573, GSE42127, GSE41271, GSE11969, and TCGA). Furthermore, we made a clear distinction in staging as it may be prognostically fairly different: Stage Ia/Ib/Ia/Ib.

Next, performance status (PS) was not included as the information on PS score was not provided in datasets we used for our validation cohort. More importantly, as our validation cohort only consists of early-stage patients, they are likely to be either ECOG 0 or 1. As the use of ECOG score is quite controversial [2] and can be subjective [3], it's usefulness in our multivariate analysis may be limited. To ensure our analysis encompassed critical disease parameters, we have included histology (ADC/LCC/SCC), gender (female/male), smoking (never/ever), and stage (IA/IB/IIA/IIB), which were provided in the annotation file of each dataset in our multivariate analyses (Table 1).

Comment: Notable in the high -risk score group were more patient with higher stage, older patient, more smokers, more female and more often adenocarcinoma patients, thus all parameters associated with poor prognosis. Although the risk score seems to be better in the multivariate Cox regression models of each single study a combination of these clinical parameters was not compared.

Response: We have included a bar graph comparing proportion of patients assigned in high- and low-risk group in terms of each clinical parameter. It may be confusing as we showed the proportion of patients in percentage. However, it

is statistically important to compare the proportion of patients between low- and high-risk group rather than between patients in the same group as not the same number of patients would fall into different clinical categories (e.g., there are more adenocarcinoma samples available in public repositories than large cell carcinoma).

We appreciate your careful observation in our findings. As you highlighted, the high-risk group had more patient with higher stage compared to low-risk group. Based on our data analysis after revision, however, we believe that the difference in the proportion of patients between low- and high risk groups is not significant enough to conclude that such difference would favor the prognostic performance of the EPPI: low-risk group~20% vs. high-risk group~32.3% of patients were at Stage II. Any bias that may arise from this difference was thus further adjusted in our multivariate analyses. In terms of other clinical variables, high-risk group had fewer female patients (low-risk group~51% vs. high-risk group~38%) and adenocarcinomas (low-risk group~82% vs. high-risk group~76%), and more smokers (low-risk group~75% vs. high-risk group~80%) and younger (<65) patients (low-risk group~50% vs. high-risk group~54%). These differences in terms of difference clinical variables between low- and high-risk group are not notable as can be seen from Fig. 3c. In order to remove any undue bias that may have been exerted, complete multivariate regression were performed to account for these clinical parameters.

Comment: The selection of both adjuvant studies GSE42127 and GSE47115 is not described sufficiently. These are retrospective cohorts not based on randomized trials, thus the selection of adjuvant chemotherapy is primarily biased and interpretation is difficult. In addition, the risk score did not clearly show the value of the ACT of high risk score patients (p=0.16, lower figure)?

Response: We greatly appreciate your valuable observations on this matter. In respond to this, we have made significant changes in evaluating the predictive

power for the ACT and covered in response to previous comments and the manuscript (page 8).

Comment: The ROC curves again showed the prognostic performance of the both data sets, already indicated in figure 3 and table 1. It does not add information on the adjuvant aspect.

Response: We would like to thank for the reviewer's careful examination. Although Figure 3 and Table 1 show the prognostic power of the EPPI, we wanted to statistically demonstrate that the EPPI has prognostic power not just at 10-year time point but also at any time period over the first 10 years after surgery. The AUC analysis results in Figure 6 further highlight that the EPPI risk score has the best prognostic accuracy in predicting survival among other traditional clinical parameters over the period of 10 years.

Comment: The discussion do not address important aspects of the field. What is the advantage of this risk score to other prognostic signatures?

Response: We truly appreciate for the insightful comments raised by the reviewer. The novelty of the EPPI besides being unique in the gene set, is the applicability to early-stage NSCLC ACT treatment decision-making. Our analysis demonstrated clear indications of early stage patients that benefitted from ACT from a random selection of patient groups. This allows patients to derive the best benefits of the ACT and minimizing the drug side effects for non-responders.

The second best option is to look at all relevant traditional clinical parameters such as gender, age, smoking history and histology of NSCLC as these greatly affect the tumor. Unlike prior studies claiming prognostic gene signatures, a thorough investigation of the prognostic performance of the EPPI in comparison with other clinical variables was statistically done with complete multivariate survival analyses and the ROC/AUC analyses.

Comment: What is the overlap of genes included in this risk score to other signatures?

Response: In respond to this, we did a comprehensive literature review on other prognostic gene signatures for NSCLC. In short, the overlap between the EPPI signature and previously reported prognostic genes was very minimal: only *CHRD1* and *SPP1* were overlapping in the gene panel reported in Chen et al. study. This demonstrated the uniqueness of our signature gene set.

None of these prior studies had significance of ECM-related genes. In fact, prognostic signatures were poorly overlapping. We hypothesized that different statistical models adopted to develop prognostic model applied to heterogeneous NSCLC in a limited validation cohort lead to the lack of concordance between prognostic signatures. This was, in fact, our main motivation to solve the problems with the 'big data' approach to increase the sample size and validate in the largest patient pool.

Moreover, this is the first study to develop prognostic gene panel exclusively for early-stage NSCLC patients who can truly avoid unnecessary risks and side

effects of chemotherapy.

Comment: What is the practical effect of the application of the risk score? Is it possible to spare patients from adjuvant therapy? Can this method be applied in clinical practice?

Response: We truly appreciate the reviewer for bringing up the matter on the practical application of the EPPI risk score in clinical settings. In the traditional clinical practice, when looking at traditional clinical factors such as tumor size, stage of cancer, or age alone, NSCLC patients may appear to have an aggressive tumor with the high risk of recurrence, which indicates potential need for chemotherapy treatment. With the use of EPPI signature, on the other hand, physicians may look directly at a tumor's underlying biology. In this regard, the same patient that was assessed clinically as high risk could be genomically low risk by EPPI, thus potentially avoid overtreatment.

The EPPI risk score was developed by analyzing the entire human genome for 20,000 genes in more than 2,000 NSCLC patients. As seen in our bioinformatics results, more than half of early-stage NSCLC patients (56.3%) could be identified as low risk by the EPPI with no meaningful benefit of adding chemotherapy, potentially sparing thousands of early-stage NSCLC patients from unnecessary treatment.

In response to the reviewer's valuable comments altogether, we have made significant improvements in demonstrating the clinical utility of the EPPI risk scoring metrics. The identification of universal cut-off score shows potential use of the GeneChip Human Genome U133 Plus 2.0 Array as a diagnostic platform in the clinical setting. For those who have gone through resection surgery, resected tumor tissue can be preserved and sent in service pack for subsequent QC check. A 29-gene DNA Affymetrix GeneChip will then be used for the 29-gene expression levels where the EPPI risk score can be computed to be

determined as either low- or high-risk. There is no intermediate and thus the EPPI can provide personalized and definitive measure in likely of survival outcomes such as overall survival expectancy and that of returning the cancer. It allows both doctor and patient in determining whether chemotherapy is needed based on provided statistics and the EPPI risk score.

As shown in our validation cohort, our target patient pool is early-stage NSCLC as they are the ones who could potentially be spared from unnecessary overtreatment. Our bioinformatics results revealed that more than half of early-stage NSCLC patients (56.3%) could be identified as low-risk by the EPPI with no meaningful benefit of adding chemotherapy.

On top of the databases that we have currently established, we intend in our future work to collect prospective local clinical samples to validate and possibly refine our cutoff criteria. The EPPI risk score may greatly assist physicians in providing a personalized, quantitative, and definitive analysis so the physicians can have a clearer overview of the probability of cancer recurrence – which is a significantly unmet need existing in the current clinical practice.

Is it feasible to perform gene expression arrays for each operated patient on fresh frozen tissue?

Response: We greatly appreciate the reviewer for the insightful query. Given the clinical benefits that can be derived, it outweighs the physical cost for such assays. Furthermore, routine molecular profiling for EGFR, ALK, ROS, etc., are already performed and the addition of these small numbers of genes do not add much complexity to sample extraction or analysis. The direct clinical benefit that can be derived is the early identification of high risk individuals with poor outcome and it addresses a critical need for clinical prescription of ACT for early-stage NSCLC.

Response to Comments from Reviewer #2 (Expert in Cancer Gene Expression and Prognosis)

Overall Comments: A. The paper presents an ECM-related prognostic and predictive indicator (EPPI), based on the expression values of 29 genes in NSCLC patients. The EPPI is generated by calculating the tumor vs normal fold change (FC) for a microarray dataset of 925 tumor and 193 normal samples (n=1118), assembled from several independent datasets. 103 differentially expressed genes (DEGs) were identified. Next, gene set enrichment analysis was performed on these 103 genes (separately on up and down regulated genes) on RNAseq data from TCGA datasets (n=1128); a group of 29 ECM-related genes constituted the largest significantly enriched "cellular component". The EPPI risk score was defined as "the geometric mean ratio of the expression values of highly to lowly expressed gene clusters". The prognostic and predictive performance of the risk score was tested on a large number of independent cohorts, using the median value of the EPPI to separate the tumor samples of each cohort into high and low risk groups. Good performance was found.

B. The literature abounds with prognostic and predictive gene lists and associated risk scores for various cancers. This paper has in my opinion three significant advantages. 1. Existing biological knowledge is used for gene selection, as opposed to blindly ranking genes on the basis of some figure of merit. 2. The number of samples is large and many independent datasets are tested, indicating robustness of the success rate of the risk score. 3. The main focus is on a particular kind of lung cancer - NSCLC.

C. The data and analysis presented in the paper are solid and of high quality.

D. The statistical analysis is professionally done.

I do have a few points of criticism and suggestions that the authors should relate to and perform.

Comment: E. Robustness of prognostic gene lists has been a central issue in evaluation of studies of this type, see Ein Dor et al Bioinformatics vol 21 p 171 (2005). The overlap between "prognostic gene sets" produced by different studies tends to be dismally low. The present study uses previous knowledge (Gene Ontologies, databases to assign genes to biological function) and hence it has a much better chance of producing a robust gene set. I strongly recommend testing this hypothesis: use the two independent datasets n=1118 and n=1128) in a similar fashion (identifying DEGs, perform enrichment analysis) to produce two most enriched gene sets: see whether one gets the same genes when this is done, or - report the overlap between the two gene lists found. This is an essential step to dispel the existing suspicions against feature selection methods based on ranking genes.

Response: We are grateful for all the comments and meaningful insights that the reviewer has provided us to further deepen enthusiasms for our findings.

As the reviewer suggested, we have performed a thorough statistical analysis to completely make certain that our final DEG list is not a result of any bias selection: the overlap between DEG lists derived from random sampling was computed for 10,000 iterations (Fig. 2d). Each iteration was done to define 'random set' and 'remaining set' and apply the same statistical criteria to identify DEG list using our initial discovery cohort (merged 10 independent datasets, 1118 samples). Mean overlap coefficient between two randomly defined cohorts for 10,000 iterations was 0.899, suggesting that our final DEG list does not carry any existing bias against feature selection method.

Comment: F1. I find the definition of the EPPI risk score ambiguous - I think I can guess what they are doing, but I would feel much more comfortable with a more careful definition and even a formula.

Response: We would like to express our gratitude for sharing your opinions on the previous EPPI risk scoring metrics. There has been a significant change in developing the EPPI risk scoring system accordingly.

1. The 'discovery cohort' was first established from initial 10 datasets where only early-stage NSCLC tumor samples with complete survival information were selected (from initial 925 primary NSCLC tumors from 10 datasets, 489 samples met these criteria).

2. The 'discovery cohort' was then used to compute the Cox regression coefficient for each EPPI gene (Supplementary Table 3, Fig. 3b). The EPPI risk score was then calculated using these computed coefficients with the gene expression values of 29 EPPI genes :

Formula = $29 + \sum (\text{Regression Coefficient}) * (\text{Gene Expression})$

Using the defined regression coefficient for each EPPI gene, we could generate personalized individual EPPI risk score with mRNA expression values of 29 EPPI genes.

Comment: F2. As I understand the manner in which the score is tested and applied, one needs an entire cohort to assign a sample to high/low risk groups (by the median over the cohort). This makes clinical application of the score rather difficult. How would setting a "universal" threshold, on the basis of the "training" phase, and using it in all tests, affect the quality of the prognostic and predictive results?

Response: In response to this, there has been a major improvement in proving potential clinical utility of the EPPI risk scoring metrics: We have optimized the cut-off score using the most commonly probed microarray platform, Affymetrix-GPL570. The optimal EPPI risk score was derived using the ROC/AUC analyses, and it has been validated in all independent datasets probed with Affymetrix-

GPL570 (Discovery set, GSE3141, GSE37745, and GSE30219): patients stratified into low- and high-risk groups had significant difference in terms of survival outcomes.

The determination of the common cut-off EPPI risk score truly shows the potential clinical utility of the EPPI. As seen in our bioinformatics results, more than half of early-stage NSCLC patients (56.3%) could be identified as low risk by the EPPI with no meaningful benefit of adding chemotherapy, potentially sparing thousands of early-stage NSCLC patients from unnecessary treatment.

The same patient that was assessed clinically as high risk could be genomically low risk by EPPI, thus potentially avoid overtreatment. Current clinical settings lack multigene diagnostic assays due to poorly overlapping prognostic genetic signatures in NSCLC. Different statistical methods applied in prior studies handle heterogeneous NSCLC differently and further lack large validation pool. Such limitation truly motivated us to use the 'big data' approach so as to increase the sample size for robust validation.

On top of the databases that we have currently established, we intend in our future work to collect prospective local clinical samples to validate and possibly refine our cutoff criteria. The EPPI risk score may greatly assist physicians in providing a personalized, quantitative, and definitive analysis so the physicians can have a clearer overview of the probability of cancer recurrence – which is a significantly unmet need existing in the current clinical practice.

Response to Comments from Reviewer #3 (Statistical Expert)

Overall Comments: Early-stage lung cancer is a disease setting where a biomarker for that would identify a subset who would benefit from adjuvant chemotherapy has been of interest for some time. Currently, patients generally don't get adjuvant chemotherapy. Several studies have attempted to do this and this current study is also addressing this problem. The authors claim to have had some success and if this is true that would be a potentially important finding. In particular, the authors focused on a extracellular matrix associated signature consisting of 29 genes.

The interest in this study is that the authors have gathered together a large number of samples' data from many different sources. They also provide some justification for the focus on ECM-associated genes.

Comment: In general, the data and methodology used are strong, although there are some aspects of the study which dampen enthusiasm for the findings. One is that, in each dataset, different cutpoints were used for the classification (based on the median signature score). The manuscript would have been greatly strengthened is some attempt was made to evaluate the potential of a common cutpoint decision rule for all patients.

Response: Firstly, we would like to show deep appreciation for your detailed comments and providing new aspects to greatly add value to our manuscript.

There has been a major improvement in the analysis to prove the potential clinical utility of the EPPI risk scoring metrics: We have optimized the cut-off score using the most commonly probed microarray platform, Affymetrix-GPL570. The optimized EPPI risk score was derived using the ROC/AUC analyses, and it

has been validated in all independent datasets probed with Affymetrix-GPL570 (Discovery set, GSE3141, GSE37745, and GSE30219): patients stratified into low- and high-risk groups had significant difference in terms of survival outcomes.

The determination of the common cut-off EPPI risk score truly shows the potential clinical utility of the EPPI. As seen in our bioinformatics results, more than half of early-stage NSCLC patients (56.3%) could be identified as low risk by the EPPI with no meaningful benefit of adding chemotherapy, potentially sparing thousands of early-stage NSCLC patients from unnecessary treatment.

The same patient that was assessed clinically as high risk could be genomically low risk by EPPI, thus potentially avoid overtreatment. Current clinical settings lack multigene diagnostic assays due to poorly overlapping prognostic genetic signatures in NSCLC. Different statistical methods applied in prior studies handle heterogeneous NSCLC differently and further lack large validation pool. Such limitation truly motivated us to use the 'big data' approach so as to increase the sample size for robust validation.

On top of the database that we have currently established, the collection of clinical local samples will further add great values in developing effective diagnostic genomic kit. The EPPI risk score may greatly assist physicians in providing a personalized, quantitative, and definitive result so the patient can know with certainty of cancer recurrence – which is a significantly unmet need existing in the current clinical practice.

Comment: A second issue is the treatment of tumor stage. It was not clear how the staging was done. First, the version of the AJCC scoring system that was used for tumor staging needs to be given, since the system evolves over time. Also, there was not a distinction made between stage IA and IB, which are prognostically fairly different, and this data is generally available and therefore

relevant for the multivariate analysis which purports to tell us whether provided information is novel.

Response: We would like to express our sincere appreciation in suggesting new aspects in the matter of tumor stage. We have made significant changes in multivariate survival analyses where all samples were clearly annotated with one of the four early-stage AJCC tumor staging: Stage IA, IB, IIA, and IIB. Instead of combining Stage IA and IB into Stage I as previous analyses, all multivariate survival analyses were performed again with clear distinction between Stage IA and IB, and Stage IIA and IIB (Table 1).

Comment: A third issue is that the patients who received ACT were not randomized to it, presumably, which complicates the interpretation of the Kaplan Meier curves in Figure 4. (Also, relatedly, why is not an interaction p-value provided here?)

Response: We understand that the limitation of our study is limited dataset available in public repositories to evaluate the predictive power for ACT. Among all datasets we used to develop the prognostic model in this study including both discovery and validation datasets, only GSE68465 and GSE42127 had information on the ACT history and thus we were able to filter patients who met our defined criteria for analyses. GSE68465, however, had information on adjuvant radiotherapy (ART) history as well. The presence of ART history may bias the results for the evaluation of the predictive power. Moreover, if we only consider patients who received ACT and not ART, only 43 patients could be selected for analysis in evaluating the predictive power for ACT. In this regard, we decided only to use GSE42127 to show the potential predictive power of the EPPI for ACT in 144 patients altogether.

Comment: The authors also state on line 194 that the implementation of the EPPI signature in the clinical setting is straightforward. This suggests a lack of

input from clinicians on the manuscript writing at least. More specifically, some discussion of the potential patient population and clinical setting this project is aiming towards seems needed.

Response: In response to this, we have made significant changes in showing the clinical utility of the EPPI risk scoring metrics. The identification of universal cut-off score shows potential use of the GeneChip Human Genome U133 Plus 2.0 Array as a diagnostic platform in the clinical setting. For those who have gone through resection surgery, resected tumor tissue can be preserved and sent in service pack for subsequent QC check. A 29-gene DNA Affymetrix GeneChip will then be used for the 29-gene expression levels where the EPPI risk score can be computed to be determined as either low- or high-risk. There is no intermediate and thus the EPPI can provide personalized and definitive measure in likely of survival outcomes such as overall survival expectancy and that of returning the cancer. It allows both doctor and patient in determining whether chemotherapy is needed based on provided statistics and the EPPI risk score.

As shown in our validation cohort, our target patient pool is early-stage NSCLC as they are the ones who could potentially be spared from unnecessary overtreatment. Our bioinformatics results revealed that more than half of early-stage NSCLC patients (56.3%) could be identified as low-risk by the EPPI with no meaningful benefit of adding chemotherapy.

On top of the database that we have currently established, we intend in future work to collect prospective local clinical samples to validate and possibly refine out cutoff criteria. The EPPI risk score may greatly assist physicians in providing a personalized, quantitative, and definitive result so the patient can know with certainty of cancer recurrence – which is a significantly unmet need existing in the current clinical practice.

Comment: In Table 1, why are there areas with missing data. This needs to be

explained. Was there a model selection step, or is the data missing in the original datasets? More generally, the level of missing covariate data in each dataset should be provided somewhere.

Response: We have now stated in Table 1 that the missing data were left to be empty in the table as the respective data were not provided in the original datasets. As our validation cohort consists of a great number of independent datasets, not all clinical parameters are provided for each dataset. However, except for GSE3141, the rest had almost complete information on traditional clinical parameters (histology, gender, smoking, AJCC staging and age).

Comment: The authors also state on line 164 that the "...best trade-off between sensitivity and specificity was ...". But the best tradeoff depends on clinical factors like prevalence, cost, side-effects, etc. Clinical context is needed for such claims.

Response: We greatly appreciate the reviewer for the insightful comment. We agree that the sensitivity and specificity of the EPPI signature will be dependent on various clinical factors and outcomes. The ROC analysis in this study provided a means to determine the initial patient scoring cutoff value, which was subsequently verified in our validation cohort using the public databases. With clinical factors included, we further plan to optimize our cutoff scores with local clinical samples in a prospectively-conducted study. The current finding using the public databases and further optimization of EPPI risk scoring metrics with our observation of clinical patients form an interesting work in future endeavors.

Thank you once again for the suggestions and we believe that all the proposed comments are addressed. We sincerely hope it merits publication in *Nature Communications*.

Yours Sincerely,

References

1. Budczies, J. et al. Cutoff Finder: A Comprehensive and Straightforward Web Application Enabling Rapid Biomarker Cutoff Optimization. *PLoS ONE* **7**, (2012).
2. Sørensen, J., Klee, M., Palshof, T. & Hansen, H. Performance status assessment in cancer patients. An inter-observer variability study. *British Journal of Cancer* **67**, 773–775 (1993).
3. Blagden, S. P., Charman, S. C., Sharples, L. D., Magee, L. R. A. & Gilligan, D. Performance status score: do patients and their oncologists agree? *British Journal of Cancer* **89**, 1022–1027 (2003)),

Reviewers' comments:

Reviewer #1 (Remarks to the Author):

The authors addressed some aspects that I raised and the manuscript improved considerably. However my main conclusion remains the same as I previously pointed out and the numerous new calculation does not compensate for the main weakness. This is still one of many studies with the same objective to identify and validate a prognostic classifier. Many of these studies include also primary data and this list is far from complete:

1. Aramburu A, Zudaire I, Pajares MJ, Agorreta J, Orta A, Lozano MD, Gúrpide A, Gómez-Román J, Martínez-Climent JA, Jassem J, Skrzypski M, Suraokar M, Behrens C, Wistuba II, Pio R, Rubio A, Montuenga LM. Combined clinical and genomic signatures for the prognosis of early stage non-small cell lung cancer based on gene copy number alterations. *BMC Genomics*. 2015 Oct 6;16:752.
2. Baty F, Facompré M, Kaiser S, Schumacher M, Pless M, Bubendorf L, Savic S, Marrer E, Budach W, Buess M, Kehren J, Tamm M, Brutsche MH. Gene profiling of clinical routine biopsies and prediction of survival in non-small cell lung cancer. *Am J Respir Crit Care Med*. 2010 Jan 15;181(2):181-8.
3. Beer DG, Kardia SL, Huang CC, Giordano TJ, Levin AM, Misek DE, Lin L, Chen G, Gharib TG, Thomas DG, Lizyness ML, Kuick R, Hayasaka S, Taylor JM, Iannettoni MD, Orringer MB, Hanash S. Gene-expression profiles predict survival of patients with lung adenocarcinoma. *Nat Med*. 2002 Aug;8(8):816-24.
4. Bianchi F, Nuciforo P, Vecchi M, Bernard L, Tizzoni L, Marchetti A, Buttitta F, Felicioni L, Nicassio F, Di Fiore PP. Survival prediction of stage I lung adenocarcinomas by expression of 10 genes. *J Clin Invest*. 2007 Nov;117(11):3436-44.
5. Boutros PC, Lau SK, Pintilie M, Liu N, Shepherd FA, Der SD, Tsao MS, Penn LZ, Jurisica I. Prognostic gene signatures for non-small-cell lung cancer. *Proc Natl Acad Sci U S A*. 2009 Feb 24;106(8):2824-8.
6. Bueno R, Hughes E, Wagner S, Gutin AS, Lanchbury JS, Zheng Y, Archer MA, Gustafson C, Jones JT, Rushton K, Saam J, Kim E, Barberis M, Wistuba I, Wenstrup J, Wallace WA, Hartman AR, Harrison DJ. Validation of a molecular and pathological model for five-year mortality risk in patients with early stage lung adenocarcinoma. *J Thorac Oncol*. 2015 Jan;10(1):67-73.
7. Buffa FM, Harris AL, West CM, Miller CJ. Large meta-analysis of multiple cancers reveals a common, compact and highly prognostic hypoxia metagene. *Br J Cancer*. 2010 Jan 19;102(2):428-35.
8. Chen DT, Hsu YL, Fulp WJ, Coppola D, Haura EB, Yeatman TJ, Cress WD. Prognostic and predictive value of a malignancy-risk gene signature in early-stage non-small cell lung cancer. *J Natl Cancer Inst*. 2011 Dec 21;103(24):1859-70.
9. Dancik GM, Theodorescu D. Robust prognostic gene expression signatures in bladder cancer and lung adenocarcinoma depend on cell cycle related genes. *PLoS One*. 2014 Jan 22;9(1):e85249.
10. De Fraipont F, Levallet G, Creveuil C, Bergot E, Beau-Faller M, Mounawar M, Richard N, Antoine M, Rouquette I, Favrot MC, Debieuvre D, Braun D, Westeel V, Quoix E, Brambilla E, Hainaut P, Moro-Sibilot D, Morin F, Milleron B, Zalcman G; Intergroupe Francophone de Cancérologie Thoracique.. An apoptosis methylation prognostic signature for early lung cancer in the IFCT-0002 trial. *Clin Cancer Res*. 2012 May 15;18(10):2976-86.
11. Der SD, Sykes J, Pintilie M, Zhu CQ, Strumpf D, Liu N, Jurisica I, Shepherd FA, Tsao MS. Validation of a histology-independent prognostic gene signature for early-stage, non-small-cell lung cancer including stage IA patients. *J Thorac Oncol*. 2014 Jan;9(1):59-64.
12. Du L, Yamamoto S, Burnette BL, Huang D, Gao K, Jamshidi N, Kuo MD. Transcriptome profiling reveals novel gene expression signatures and regulating transcription factors of TGFβ-induced epithelial-to-mesenchymal transition. *Cancer Med*. 2016 Aug;5(8):1962-72.
13. Guo NL, Wan YW, Bose S, Denvir J, Kashon ML, Andrew ME. A novel network model identified a 13-gene lung cancer prognostic signature. *Int J Comput Biol Drug Des*. 2011;4(1):19-39
14. Guo NL, Wan YW, Tosun K, Lin H, Msiska Z, Flynn DC, Remick SC, Vallyathan V, Dowlati A, Shi

- X, Castranova V, Beer DG, Qian Y. Confirmation of gene expression-based prediction of survival in non-small cell lung cancer. *Clin Cancer Res.* 2008 Dec 15;14(24):8213-20.
15. Guo NL, Wan YW. Pathway-based identification of a smoking associated 6-gene signature predictive of lung cancer risk and survival. *Artif Intell Med.* 2012 Jun;55(2):97-105.
 16. Hou J, Aerts J, den Hamer B, van Ijcken W, den Bakker M, Riegman P, van der Leest C, van der Spek P, Foekens JA, Hoogsteden HC, Grosveld F, Philipsen S. Gene expression-based classification of non-small cell lung carcinomas and survival prediction. *PLoS One.* 2010 Apr 22;5(4):e10312.
 17. Hsu YC, Yuan S, Chen HY, Yu SL, Liu CH, Hsu PY, Wu G, Lin CH, Chang GC, Li KC, Yang PC. A four-gene signature from NCI-60 cell line for survival prediction in non-small cell lung cancer. *Clin Cancer Res.* 2009 Dec 1;15(23):7309-15.
 18. Huang S, Reitze NJ, Ewing AL, McCreary S, Uihlein AH, Brower SL, Wang D, Wang T, Gabrin MJ, Keating KE, Mulligan J, Wilson C, Davison T, McKenzie S, Tsao MS, Shepherd FA, Plamadeala V. Analytical Performance of a 15-Gene Prognostic Assay for Early-Stage Non-Small-Cell Lung Carcinoma Using RNA-Stabilized Tissue. *J Mol Diagn.* 2015 Jul;17(4):438-45.
 19. Ikehara M, Oshita F, Sekiyama A, Hamanaka N, Saito H, Yamada K, Noda K, Kameda Y, Miyagi Y. Genome-wide cDNA microarray screening to correlate gene expression profile with survival in patients with advanced lung cancer. *Oncol Rep.* 2004 May;11(5):1041-4.
 20. Kadara H, Behrens C, Yuan P, Solis L, Liu D, Gu X, Minna JD, Lee JJ, Kim E, Hong WK, Wistuba II, Lotan R. A five-gene and corresponding protein signature for stage-I lung adenocarcinoma prognosis. *Clin Cancer Res.* 2011 Mar 15;17(6):1490-501.
 21. Khirade MF, Lal G, Bapat SA. Derivation of a fifteen gene prognostic panel for six cancers. *Sci Rep.* 2015 Aug 14;5:13248.
 22. Kratz JR, Mann MJ, Jablons DM. International trial of adjuvant therapy in high risk stage I non-squamous cell carcinoma identified by a 14-gene prognostic signature. *Transl Lung Cancer Res.* 2013 Jun;2(3):222-5.
 23. Krzystanek M, Moldvay J, Szüts D, Szallasi Z, Eklund AC. A robust prognostic gene expression signature for early stage lung adenocarcinoma. *Biomark Res.* 2016 Feb 19;4:4
 24. Larsen JE, Pavey SJ, Passmore LH, Bowman R, Clarke BE, Hayward NK, Fong KM. Expression profiling defines a recurrence signature in lung squamous cell carcinoma. *Carcinogenesis.* 2007 Mar;28(3):760-6.
 25. Larsen JE, Pavey SJ, Passmore LH, Bowman RV, Hayward NK, Fong KM. Gene expression signature predicts recurrence in lung adenocarcinoma. *Clin Cancer Res.* 2007 May 15;13(10):2946-54.
 26. Lu TP, Chuang EY, Chen JJ. Identification of reproducible gene expression signatures in lung adenocarcinoma. *BMC Bioinformatics.* 2013 Dec 26;14:371.
 27. Lu Y, Wang L, Liu P, Yang P, You M. Gene-expression signature predicts postoperative recurrence in stage I non-small cell lung cancer patients. *PLoS One.* 2012;7(1):e30880.
 28. MacDermed DM, Khodarev NN, Pitroda SP, Edwards DC, Pelizzari CA, Huang L, Kufe DW, Weichselbaum RR. MUC1-associated proliferation signature predicts outcomes in lung adenocarcinoma patients. *BMC Med Genomics.* 2010 May 6;3:16.
 29. Mettu RK, Wan YW, Habermann JK, Ried T, Guo NL. A 12-gene genomic instability signature predicts clinical outcomes in multiple cancer types. *Int J Biol Markers.* 2010 Oct-Dec;25(4):219-28.
 30. Navab R, Strumpf D, Bandarchi B, Zhu CQ, Pintilie M, Ramnarine VR, Ibrahimov E, Radulovich N, Leung L, Barczyk M, Panchal D, To C, Yun JJ, Der S, Shepherd FA, Jurisica I, Tsao MS. Prognostic gene-expression signature of carcinoma-associated fibroblasts in non-small cell lung cancer. *Proc Natl Acad Sci U S A.* 2011 Apr 26;108(17):7160-5.
 31. Park YY, Park ES, Kim SB, Kim SC, Sohn BH, Chu IS, Jeong W, Mills GB, Byers LA, Lee JS. Development and validation of a prognostic gene-expression signature for lung adenocarcinoma. *PLoS One.* 2012;7(9):e44225.
 32. Pio R, Agorreta J, Montuenga LM. Prognostic signature of early lung adenocarcinoma based on the expression of ribonucleic acid metabolism-related genes. *J Thorac Cardiovasc Surg.* 2015 Oct;150(4):986-92.e1-11.
 33. Pitroda SP, Zhou T, Sweis RF, Filippo M, Labay E, Beckett MA, Mauceri HJ, Liang H, Darga TE,

- Perakis S, Khan SA, Sutton HG, Zhang W, Khodarev NN, Garcia JG, Weichselbaum RR. Tumor endothelial inflammation predicts clinical outcome in diverse human cancers. *PLoS One*. 2012;7(10):e46104.
34. Rajski M, Saaf A, Buess M. BMP2 response pattern in human lung fibroblasts predicts outcome in lung adenocarcinomas. *BMC Med Genomics*. 2015 Apr 29;8:16.
35. Raponi M, Zhang Y, Yu J, Chen G, Lee G, Taylor JM, Macdonald J, Thomas D, Moskaluk C, Wang Y, Beer DG. Gene expression signatures for predicting prognosis of squamous cell and adenocarcinomas of the lung. *Cancer Res*. 2006 Aug 1;66(15):7466-72.
36. Sanmartín E, Sirera R, Usó M, Blasco A, Gallach S, Figueroa S, Martínez N, Hernando C, Honguero A, Martorell M, Guijarro R, Rosell R, Jantus-Lewintre E, Camps C. A gene signature combining the tissue expression of three angiogenic factors is a prognostic marker in early-stage non-small cell lung cancer. *Ann Surg Oncol*. 2014 Feb;21(2):612-20.
37. Seike M, Yanaihara N, Bowman ED, Zanetti KA, Budhu A, Kumamoto K, Mechanic LE, Matsumoto S, Yokota J, Shibata T, Sugimura H, Gemma A, Kudoh S, Wang XW, Harris CC. Use of a cytokine gene expression signature in lung adenocarcinoma and the surrounding tissue as a prognostic classifier. *J Natl Cancer Inst*. 2007 Aug 15;99(16):1257-69.
38. Shahid M, Choi TG, Nguyen MN, Matondo A, Jo YH, Yoo JY, Nguyen NN, Yun HR, J, Akter S, Kang I, Ha J, Maeng CH, Kim SY, Lee JS, Kim J, Kim SS. An 8-gene signature for prediction of prognosis and chemoresponse in non-small cell lung cancer. *Oncotarget*. 2016 Dec 27;7(52):86561-86572.
39. Shibata T, Uryu S, Kokubu A, Hosoda F, Ohki M, Sakiyama T, Matsuno Y, Tsuchiya R, Kanai Y, Kondo T, Imoto I, Inazawa J, Hirohashi S. Genetic classification of lung adenocarcinoma based on array-based comparative genomic hybridization analysis: its association with clinicopathologic features. *Clin Cancer Res*. 2005 Sep 1;11(17):6177-85.
40. Shukla S, Evans JR, Malik R, Feng FY, Dhanasekaran SM, Cao X, Chen G, Beer DG, Jiang H, Chinnaiyan AM. Development of a RNA-Seq Based Prognostic Signature in Lung Adenocarcinoma. *J Natl Cancer Inst*. 2016 Oct 5;109(1).
41. Skrzypski M, Jassem E, Taron M, Sanchez JJ, Mendez P, Rzyman W, Gulida G, Raz D, Jablons D, Provencio M, Massuti B, Chaib I, Perez-Roca L, Jassem J, Rosell R. Three-gene expression signature predicts survival in early-stage squamous cell carcinoma of the lung. *Clin Cancer Res*. 2008 Aug 1;14(15):4794-9.
42. Starmans MH, Lieuwes NG, Span PN, Haider S, Dubois L, Nguyen F, van Laarhoven HW, Sweep FC, Wouters BG, Boutros PC, Lambin P. Independent and functional validation of a multi-tumour-type proliferation signature. *Br J Cancer*. 2012 Jul 24;107(3):508-15.
43. Sun Z, Wigle DA, Yang P. Non-overlapping and non-cell-type-specific gene expression signatures predict lung cancer survival. *J Clin Oncol*. 2008 Feb 20;26(6):877-83.
44. Tang H, Xiao G, Behrens C, Schiller J, Allen J, Chow CW, Suraokar M, Corvalan A, Mao J, White MA, Wistuba II, Minna JD, Xie Y. A 12-gene set predicts survival benefits from adjuvant chemotherapy in non-small cell lung cancer patients. *Clin Cancer Res*. 2013 Mar 15;19(6):1577-86.
45. Wan YW, Qian Y, Rathnagiriswaran S, Castranova V, Guo NL. A breast cancer prognostic signature predicts clinical outcomes in multiple tumor types. *Oncol Rep*. 2010 Aug;24(2):489-94.
46. Wang X, Lu Y, Feng W, Chen Q, Guo H, Sun X, Bao Y. A two kinase-gene signature model using CDK2 and PAK4 expression predicts poor outcome in non-small cell lung cancers. *Neoplasma*. 2016;63(2):322-9. 26774155.
47. Wistuba II, Behrens C, Lombardi F, Wagner S, Fujimoto J, Raso MG, Spaggiari L, Galetta D, Riley R, Hughes E, Reid J, Sangale Z, Swisher SG, Kalhor N, Moran CA, Gutin A, Lanchbury JS, Barberis M, Kim ES. Validation of a proliferation-based expression signature as prognostic marker in early stage lung adenocarcinoma. *Clin Cancer Res*. 2013 Nov 15;19(22):6261-71.
48. Wong KM, Ding K, Li S, Bradbury P, Tsao MS, Der SD, Shepherd FA, Chung C, Ng R, Seymour L, Leighl NB. A Cost-Effectiveness Analysis of Using the JBR.10-Based 15-Gene Expression Signature to Guide Adjuvant Chemotherapy in Early Stage Non-Small-Cell Lung Cancer. *Clin Lung Cancer*. 2017 Jan;18(1):e41-e47.
49. Woodard GA, Gubens MA, Jahan TM, Jones KD, Kukreja J, Theodore PR, Cardozo S, Jew G, Clary-Macy C, Jablons DM, Mann MJ. Prognostic molecular assay might improve identification of

- patients at risk for recurrence in early-stage non-small-cell lung cancer. *Clin Lung Cancer*. 2014 Nov;15(6):426-32.
50. Wu C, Zhang D. Identification of early-stage lung adenocarcinoma prognostic signatures based on statistical modeling. *Cancer Biomark*. 2017;18(2):117-123.
51. Xia R, Chen S, Chen Y, Zhang W, Zhu R, Deng A. A chromosomal passenger complex protein signature model predicts poor prognosis for non-small-cell lung cancer. *Onco Targets Ther*. 2015 Apr 7;8:721-6.
52. Xie Y, Xiao G, Coombes KR, Behrens C, Solis LM, Raso G, Girard L, Erickson HS, Roth J, Heymach JV, Moran C, Danenberg K, Minna JD, Wistuba II. Robust gene expression signature from formalin-fixed paraffin-embedded samples predicts prognosis of non-small-cell lung cancer patients. *Clin Cancer Res*. 2011 Sep 1;17(17):5705-14.
53. Xu W, Banerji S, Davie JR, Kassie F, Yee D, Kratzke R. Yin Yang gene expression ratio signature for lung cancer prognosis. *PLoS One*. 2013 Jul 17;8(7):e68742.
54. Yang X, Li H, Regan K, Li J, Huang Y, Lussier YA. Towards mechanism classifiers: expression-anchored Gene Ontology signature predicts clinical outcome in lung adenocarcinoma patients. *AMIA Annu Symp Proc*. 2012;2012:1040-9.
55. Zhou T, Wang T, Garcia JG. Expression of nicotinamide phosphoribosyltransferase-influenced genes predicts recurrence-free survival in lung and breast cancers. *Sci Rep*. 2014 Aug 22;4:6107.
56. Zhu CQ, Ding K, Strumpf D, Weir BA, Meyerson M, Pennell N, Thomas RK, Naoki K, Ladd-Acosta C, Liu N, Pintilie M, Der S, Seymour L, Jurisica I, Shepherd FA, Tsao MS. Prognostic and predictive gene signature for adjuvant chemotherapy in resected non-small-cell lung cancer. *J Clin Oncol*. 2010 Oct 10;28(29):4417-24.
57. Zhu CQ, Strumpf D, Li CY, Li Q, Liu N, Der S, Shepherd FA, Tsao MS, Jurisica I. Prognostic gene expression signature for squamous cell carcinoma of lung. *Clin Cancer Res*. 2010 Oct 15;16(20):5038-47.

This list is surely not complet but illustrates impressively that it is possible to construct a gene signatures of different genes and subsequent different biological motives, e.g. proliferation, cytokines, genomic instability, etc. (Subramanian J, *J Natl Cancer Inst*. 2010) Also several studies addressing the tumor microenvironment in lung cancer are published:

1. Roepman P, Jassem J, Smit EF, Muley T, Niklinski J, van de Velde T, Witteveen AT, Rzyman W, Floore A, Burgers S, Giaccone G, Meister M, Dienemann H, Skrzypski M, Kozlowski M, Mooi WJ, van Zandwijk N. An immune response enriched 72-gene prognostic profile for early-stage non-small-cell lung cancer. *Clin Cancer Res*. 2009 Jan 1;15(1):284-90.
2. Navab R, Strumpf D, Bandarchi B, Zhu CQ, Pintilie M, Ramnarine VR, Ibrahimov E, Radulovich N, Leung L, Barczyk M, Panchal D, To C, Yun JJ, Der S, Shepherd FA, Jurisica I, Tsao MS. Prognostic gene-expression signature of carcinoma-associated fibroblasts in non-small cell lung cancer. *Proc Natl Acad Sci U S A*. 2011 Apr 26;108(17):7160-5.
3. Du L, Yamamoto S, Burnette BL, Huang D, Gao K, Jamshidi N, Kuo MD. Transcriptome profiling reveals novel gene expression signatures and regulating transcription factors of TGFβ-induced epithelial-to-mesenchymal transition. *Cancer Med*. 2016 Aug;5(8):1962-72.
4. Chen JL, Espinosa I, Lin AY, Liao OY, van de Rijn M, West RB. Stromal responses among common carcinomas correlated with clinicopathologic features. *Clin Cancer Res*. 2013 Sep 15;19(18):5127-35.

That the genes of the presented signatures are not completely overlapping is not surprising and not new. It is also nothing special to include only stage I and II patients, most of the studies do that, in addition they show there impact in stage I or Ia patients only, e.g. Wistuba et al. *Clin Cancer Res*, 2013; Kratz et al., 2012 *JAMA*). I miss this demonstration in the present study: The challenge how to transfer the classifier to clinical practice, i.e. on small FFPE tissue biopsies has not been addressed. Indeed there are two commercial providers for lung cancer gene expression signatures (Myriad, My Plan, lung cancer and Pervenio™ Lung RS of Life Technologies) that are applicable on paraffin embedded tissue (both not cited and discussed by the authors). That these signatures are not based on ECM is for the purpose irrelevant.

Finally they did not compare their model to strong clinicopathological factors (age, stage, performance status) or factors with minor prognostic impact (smoking status, sex, histology). A multivariate analysis is not sufficient and not the same. A clinical model must be compared directly to the gene signature (Subramanian J, J Natl Cancer Inst. 2010; Subramanian et al., Nature Rev Clin Oncol). Unfortunately often publically available data bases are not sufficiently annotated for this purpose. This fact does not excuse the absence of such necessary head to head comparison.

Reviewer #2 (Remarks to the Author):

My comments and criticisms were addressed by the authors. I will go through the points raised and the authors' responses in the order listed by the response letter.

Robustness of the prognostic gene list:

1. I suggested to do a single shot double derivation of the prognostic gene list, using the same method, on the basis of two completely independent cohorts (discovery, n=1118 and validation, n=1128). The authors opted for a different approach; using the discovery set only, they partitioned it into two disjoint cohorts, X and Y, calculated the gene lists for both and determined the overlap. This was repeated 10,000 times and the resulting overlap distribution was presented. Their approach has two significant advantages over my suggestion – instead of a single one shot test of overlap they have 10,000 “events”, and they kept discovery for discovery and validation – for validation, rather than mixing the two. Their approach has a disadvantage though – the different partitions are not independent, in fact any two partitions have on the average a 50% identity of patients in the respective X and Y sets. I am willing to accept their choice.

2. Definition of the EPPI risk score:

As I understand they have changed the risk score and their current score is much clearer than the previous one. I cannot say whether this is due to better phrasing of what was done or a more transparent procedure.

3. Universal threshold:

The authors took a few very significant tests in this direction. I still miss (possibly due to my own oversight) a clear “bottom line”: say I chose the optimal threshold and working point – the resulting classifier assigns patients to high/low risk groups. How good is the resulting predictor (in terms of sensitivity and specificity)? What is the risk of not giving chemotherapy to a patient who would benefit from it? This one number (evaluated at the working parameters of the predictor) is probably the most important one, together with the complementing figure (the number or fraction of patients who do not need chemo and indeed are advised not to take it? A clear statement on this would help physicians to relate to the paper's findings.

Reviewer #3 (Remarks to the Author):

The authors have addressed my concerns very well.

Responses to Reviewers' Comments

We would like to thank the reviewers for the time and effort in reviewing our manuscript entitled "Integrative genomic approach identifies a 29-gene extracellular matrix-related prognostic and predictive indicator (EPPI) for early-stage non-small cell lung cancer". Below are the queries raised with point by point response from us. We sincerely hope that this will sufficiently address all the concerns raised.

Responses to Comments from Reviewer #1

Comment: The authors addressed some aspects that I raised and the manuscript improved considerably. However, my main conclusion remains the same as I previously pointed out and the numerous new calculation does not compensate for the main weakness. This is still one of many studies with the same objective to identify and validate a prognostic classifier. Many of these studies include also primary data and this list is far from complete:

1. Aramburu A, Zudaire I, Pajares MJ, Agorreta J, Orta A, Lozano MD, Gúrpide A, Gómez-Román J, Martínez-Climent JA, Jassem J, Skrzypski M, Suraokar M, Behrens C, Wistuba II, Pio R, Rubio A, Montuenga LM. Combined clinical and genomic signatures for the prognosis of early stage non-small cell lung cancer based on gene copy number alterations. *BMC Genomics*. 2015 Oct 6;16:752.
2. Baty F, Facompré M, Kaiser S, Schumacher M, Pless M, Bubendorf L, Savic S, Marrer E, Budach W, Buess M, Kehren J, Tamm M, Brutsche MH. Gene profiling of clinical routine biopsies and prediction of survival in non-small cell lung cancer. *Am J Respir Crit Care Med*. 2010 Jan 15;181(2):181-8.
3. Beer DG, Kardia SL, Huang CC, Giordano TJ, Levin AM, Misek DE, Lin L, Chen G, Gharib TG, Thomas DG, Lizyness ML, Kuick R, Hayasaka S, Taylor JM, Iannettoni MD, Orringer MB, Hanash S. Gene-expression profiles predict survival of patients with lung adenocarcinoma. *Nat Med*. 2002 Aug;8(8):816-24.
4. Bianchi F, Nuciforo P, Vecchi M, Bernard L, Tizzoni L, Marchetti A, Buttitta F, Felicioni L, Nicassio F, Di Fiore PP. Survival prediction of stage I lung adenocarcinomas by expression of 10 genes. *J Clin Invest*. 2007 Nov;117(11):3436-44.
5. Boutros PC, Lau SK, Pintilie M, Liu N, Shepherd FA, Der SD, Tsao MS, Penn LZ, Jurisica I. Prognostic gene signatures for non-small-cell lung cancer. *Proc Natl Acad Sci U S A*. 2009 Feb 24;106(8):2824-8.
6. Bueno R, Hughes E, Wagner S, Gutin AS, Lanchbury JS, Zheng Y, Archer MA, Gustafson C, Jones JT, Rushton K, Saam J, Kim E, Barberis M, Wistuba I, Wenstrup J, Wallace WA, Hartman AR, Harrison DJ. Validation of a molecular and pathological model for five-year mortality risk in patients with early stage lung adenocarcinoma. *J Thorac Oncol*. 2015 Jan;10(1):67-73.
7. Buffa FM, Harris AL, West CM, Miller CJ. Large meta-analysis of multiple cancers reveals a common, compact and highly prognostic hypoxia metagene. *Br J Cancer*. 2010 Jan 19;102(2):428-35.
8. Chen DT, Hsu YL, Fulp WJ, Coppola D, Haura EB, Yeatman TJ, Cress WD. Prognostic and predictive value of a malignancy-risk gene signature in early-stage non-small cell lung cancer. *J Natl Cancer Inst*. 2011 Dec 21;103(24):1859-70.
9. Dancik GM, Theodorescu D. Robust prognostic gene expression signatures in bladder cancer and lung adenocarcinoma depend on cell cycle related genes. *PLoS One*. 2014 Jan 22;9(1):e85249.
10. De Fraipont F, Levallet G, Creveuil C, Bergot E, Beau-Faller M, Mounawar M, Richard N, Antoine M, Rouquette I, Favrot MC, Debieuvre D, Braun D, Westeel V, Quoix E, Brambilla E, Hainaut P, Moro-Sibilot D, Morin F, Milleron B, Zalcman G; Intergroupe Francophone de Cancérologie Thoracique.. An apoptosis methylation prognostic signature for early lung cancer in the IFCT-0002 trial. *Clin Cancer Res*. 2012 May 15;18(10):2976-86.
11. Der SD, Sykes J, Pintilie M, Zhu CQ, Strumpf D, Liu N, Jurisica I, Shepherd FA, Tsao MS. Validation of a histology-independent prognostic gene signature for early-stage, non-small-cell lung cancer including stage IA patients. *J Thorac Oncol*. 2014 Jan;9(1):59-64.
12. Du L, Yamamoto S, Burnette BL, Huang D, Gao K, Jamshidi N, Kuo MD. Transcriptome profiling reveals novel gene expression signatures and regulating transcription factors of TGFβ-induced epithelial-to-mesenchymal transition. *Cancer Med*. 2016 Aug;5(8):1962-72.
13. Guo NL, Wan YW, Bose S, Denvir J, Kashon ML, Andrew ME. A novel network model identified a 13-gene lung

cancer prognostic signature. *Int J Comput Biol Drug Des.* 2011;4(1):19-39

14. Guo NL, Wan YW, Tosun K, Lin H, Msiska Z, Flynn DC, Remick SC, Vallyathan V, Dowlati A, Shi X, Castranova V, Beer DG, Qian Y. Confirmation of gene expression-based prediction of survival in non-small cell lung cancer. *Clin Cancer Res.* 2008 Dec 15;14(24):8213-20.

15. Guo NL, Wan YW. Pathway-based identification of a smoking associated 6-gene signature predictive of lung cancer risk and survival. *Artif Intell Med.* 2012 Jun;55(2):97-105.

16. Hou J, Aerts J, den Hamer B, van Ijcken W, den Bakker M, Riegman P, van der Leest C, van der Spek P, Foekens JA, Hoogsteden HC, Grosveld F, Philipsen S. Gene expression-based classification of non-small cell lung carcinomas and survival prediction. *PLoS One.* 2010 Apr 22;5(4):e10312.

17. Hsu YC, Yuan S, Chen HY, Yu SL, Liu CH, Hsu PY, Wu G, Lin CH, Chang GC, Li KC, Yang PC. A four-gene signature from NCI-60 cell line for survival prediction in non-small cell lung cancer. *Clin Cancer Res.* 2009 Dec 1;15(23):7309-15.

18. Huang S, Reitze NJ, Ewing AL, McCreary S, Uihlein AH, Brower SL, Wang D, Wang T, Gabrin MJ, Keating KE, Mulligan J, Wilson C, Davison T, McKenzie S, Tsao MS, Shepherd FA, Plamadeala V. Analytical Performance of a 15-Gene Prognostic Assay for Early-Stage Non-Small-Cell Lung Carcinoma Using RNA-Stabilized Tissue. *J Mol Diagn.* 2015 Jul;17(4):438-45.

19. Ikehara M, Oshita F, Sekiyama A, Hamanaka N, Saito H, Yamada K, Noda K, Kameda Y, Miyagi Y. Genome-wide cDNA microarray screening to correlate gene expression profile with survival in patients with advanced lung cancer. *Oncol Rep.* 2004 May;11(5):1041-4.

20. Kadara H, Behrens C, Yuan P, Solis L, Liu D, Gu X, Minna JD, Lee JJ, Kim E, Hong WK, Wistuba II, Lotan R. A five-gene and corresponding protein signature for stage-I lung adenocarcinoma prognosis. *Clin Cancer Res.* 2011 Mar 15;17(6):1490-501.

21. Khirade MF, Lal G, Bapat SA. Derivation of a fifteen gene prognostic panel for six cancers. *Sci Rep.* 2015 Aug 14;5:13248.

22. Kratz JR, Mann MJ, Jablons DM. International trial of adjuvant therapy in high risk stage I non-squamous cell carcinoma identified by a 14-gene prognostic signature. *Transl Lung Cancer Res.* 2013 Jun;2(1):222-5.

23. Krzystanek M, Moldvay J, Szüts D, Szallasi Z, Eklund AC. A robust prognostic gene expression signature for early stage lung adenocarcinoma. *Biomark Res.* 2016 Feb 19;4:4

24. Larsen JE, Pavey SJ, Passmore LH, Bowman R, Clarke BE, Hayward NK, Fong KM. Expression profiling defines a recurrence signature in lung squamous cell carcinoma. *Carcinogenesis.* 2007 Mar;28(1):760-6.

25. Larsen JE, Pavey SJ, Passmore LH, Bowman RV, Hayward NK, Fong KM. Gene expression signature predicts recurrence in lung adenocarcinoma. *Clin Cancer Res.* 2007 May 15;13(10):2946-54.

26. Lu TP, Chuang EY, Chen JJ. Identification of reproducible gene expression signatures in lung adenocarcinoma. *BMC Bioinformatics.* 2013 Dec 26;14:371.

27. Lu Y, Wang L, Liu P, Yang P, You M. Gene-expression signature predicts postoperative recurrence in stage I non-small cell lung cancer patients. *PLoS One.* 2012;7(1):e30880.

28. MacDermed DM, Khodarev NN, Pitroda SP, Edwards DC, Pelizzari CA, Huang L, Kufe DW, Weichselbaum RR. MUC1-associated proliferation signature predicts outcomes in lung adenocarcinoma patients. *BMC Med Genomics.* 2010 May 6;3:16.

29. Mettu RK, Wan YW, Habermann JK, Ried T, Guo NL. A 12-gene genomic instability signature predicts clinical outcomes in multiple cancer types. *Int J Biol Markers.* 2010 Oct-Dec;25(4):219-28.

30. Navab R, Strumpf D, Bandarchi B, Zhu CQ, Pintilie M, Ramnarine VR, Ibrahimov E, Radulovich N, Leung L, Barczyk M, Panchal D, To C, Yun JJ, Der S, Shepherd FA, Jurisica I, Tsao MS. Prognostic gene-expression signature of carcinoma-associated fibroblasts in non-small cell lung cancer. *Proc Natl Acad Sci U S A.* 2011 Apr 26;108(17):7160-5.

31. Park YY, Park ES, Kim SB, Kim SC, Sohn BH, Chu IS, Jeong W, Mills GB, Byers LA, Lee JS. Development and validation of a prognostic gene-expression signature for lung adenocarcinoma. *PLoS One.* 2012;7(9):e44225.

32. Pio R, Agorreta J, Montuenga LM. Prognostic signature of early lung adenocarcinoma based on the expression of ribonucleic acid metabolism-related genes. *J Thorac Cardiovasc Surg.* 2015 Oct;150(4):986-92.e1-11.

33. Pitroda SP, Zhou T, Sweis RF, Filippo M, Labay E, Becktt MA, Mauceri HJ, Liang H, Darga TE, Perakis S, Khan SA, Sutton HG, Zhang W, Khodarev NN, Garcia JG, Weichselbaum RR. Tumor endothelial inflammation predicts clinical outcome in diverse human cancers. *PLoS One.* 2012;7(10):e46104.

34. Rajski M, Saaf A, Buess M. BMP2 response pattern in human lung fibroblasts predicts outcome in lung adenocarcinomas. *BMC Med Genomics.* 2015 Apr 29;8:16.

35. Raponi M, Zhang Y, Yu J, Chen G, Lee G, Taylor JM, Macdonald J, Thomas D, Moskaluk C, Wang Y, Beer DG. Gene expression signatures for predicting prognosis of squamous cell and adenocarcinomas of the lung. *Cancer Res.* 2006 Aug 1;66(15):7466-72.

36. Sanmartín E, Sirera R, Usó M, Blasco A, Gallach S, Figueroa S, Martínez N, Hernando C, Honguero A, Martorell M, Guijarro R, Rosell R, Jantus-Lewintre E, Camps C. A gene signature combining the tissue expression of three angiogenic factors is a prognostic marker in early-stage non-small cell lung cancer. *Ann Surg Oncol.* 2014 Feb;21(2):612-20.

37. Seike M, Yanaihara N, Bowman ED, Zanetti KA, Budhu A, Kumamoto K, Mechanic LE, Matsumoto S, Yokota J, Shibata T, Sugimura H, Gemma A, Kudoh S, Wang XW, Harris CC. Use of a cytokine gene expression signature in lung adenocarcinoma and the surrounding tissue as a prognostic classifier. *J Natl Cancer Inst.* 2007 Aug 15;99(16):1257-69.
38. Shahid M, Choi TG, Nguyen MN, Matondo A, Jo YH, Yoo JY, Nguyen NN, Yun HR, J, Akter S, Kang I, Ha J, Maeng CH, Kim SY, Lee JS, Kim J, Kim SS. An 8-gene signature for prediction of prognosis and chemoresponse in non-small cell lung cancer. *Oncotarget.* 2016 Dec 27;7(52):86561-86572.
39. Shibata T, Uryu S, Kokubu A, Hosoda F, Ohki M, Sakiyama T, Matsuno Y, Tsuchiya R, Kanai Y, Kondo T, Imoto I, Inazawa J, Hirohashi S. Genetic classification of lung adenocarcinoma based on array-based comparative genomic hybridization analysis: its association with clinicopathologic features. *Clin Cancer Res.* 2005 Sep 1;11(17):6177-85.
40. Shukla S, Evans JR, Malik R, Feng FY, Dhanasekaran SM, Cao X, Chen G, Beer DG, Jiang H, Chinnaiyan AM. Development of a RNA-Seq Based Prognostic Signature in Lung Adenocarcinoma. *J Natl Cancer Inst.* 2016 Oct 5;109(1).
41. Skrzypski M, Jassem E, Taron M, Sanchez JJ, Mendez P, Rzyman W, Gulida G, Raz D, Jablons D, Provencio M, Massuti B, Chaib I, Perez-Roca L, Jassem J, Rosell R. Three-gene expression signature predicts survival in early-stage squamous cell carcinoma of the lung. *Clin Cancer Res.* 2008 Aug 1;14(15):4794-9.
42. Starmans MH, Lieuwes NG, Span PN, Haider S, Dubois L, Nguyen F, van Laarhoven HW, Sweep FC, Wouters BG, Boutros PC, Lambin P. Independent and functional validation of a multi-tumour-type proliferation signature. *Br J Cancer.* 2012 Jul 24;107(1):508-15.
43. Sun Z, Wigle DA, Yang P. Non-overlapping and non-cell-type-specific gene expression signatures predict lung cancer survival. *J Clin Oncol.* 2008 Feb 20;26(6):877-83.
44. Tang H, Xiao G, Behrens C, Schiller J, Allen J, Chow CW, Suraokar M, Corvalan A, Mao J, White MA, Wistuba II, Minna JD, Xie Y. A 12-gene set predicts survival benefits from adjuvant chemotherapy in non-small cell lung cancer patients. *Clin Cancer Res.* 2013 Mar 15;19(6):1577-86.
45. Wan YW, Qian Y, Rathnagiriswaran S, Castranova V, Guo NL. A breast cancer prognostic signature predicts clinical outcomes in multiple tumor types. *Oncol Rep.* 2010 Aug;24(2):489-94.
46. Wang X, Lu Y, Feng W, Chen Q, Guo H, Sun X, Bao Y. A two kinase-gene signature model using CDK2 and PAK4 expression predicts poor outcome in non-small cell lung cancers. *Neoplasma.* 2016;63(2):322-9. 26774155.
47. Wistuba II, Behrens C, Lombardi F, Wagner S, Fujimoto J, Raso MG, Spaggiari L, Galetta D, Riley R, Hughes E, Reid J, Sangale Z, Swisher SG, Kalhor N, Moran CA, Gutin A, Lanchbury JS, Barberis M, Kim ES. Validation of a proliferation-based expression signature as prognostic marker in early stage lung adenocarcinoma. *Clin Cancer Res.* 2013 Nov 15;19(22):6261-71.
48. Wong KM, Ding K, Li S, Bradbury P, Tsao MS, Der SD, Shepherd FA, Chung C, Ng R, Seymour L, Leighl NB. A Cost-Effectiveness Analysis of Using the JBR.10-Based 15-Gene Expression Signature to Guide Adjuvant Chemotherapy in Early Stage Non-Small-Cell Lung Cancer. *Clin Lung Cancer.* 2017 Jan;18(1):e41-e47.
49. Woodard GA, Gubens MA, Jahan TM, Jones KD, Kukreja J, Theodore PR, Cardozo S, Jew G, Clary-Macy C, Jablons DM, Mann MJ. Prognostic molecular assay might improve identification of patients at risk for recurrence in early-stage non-small-cell lung cancer. *Clin Lung Cancer.* 2014 Nov;15(6):426-32.
50. Wu C, Zhang D. Identification of early-stage lung adenocarcinoma prognostic signatures based on statistical modeling. *Cancer Biomark.* 2017;18(2):117-123.
51. Xia R, Chen S, Chen Y, Zhang W, Zhu R, Deng A. A chromosomal passenger complex protein signature model predicts poor prognosis for non-small-cell lung cancer. *Onco Targets Ther.* 2015 Apr 7;8:721-6.
52. Xie Y, Xiao G, Coombes KR, Behrens C, Solis LM, Raso G, Girard L, Erickson HS, Roth J, Heymach JV, Moran C, Danenberg K, Minna JD, Wistuba II. Robust gene expression signature from formalin-fixed paraffin-embedded samples predicts prognosis of non-small-cell lung cancer patients. *Clin Cancer Res.* 2011 Sep 1;17(17):5705-14.
53. Xu W, Banerji S, Davie JR, Kassie F, Yee D, Kratzke R. Yin Yang gene expression ratio signature for lung cancer prognosis. *PLoS One.* 2013 Jul 17;8(7):e68742.
54. Yang X, Li H, Regan K, Li J, Huang Y, Lussier YA. Towards mechanism classifiers: expression-anchored Gene Ontology signature predicts clinical outcome in lung adenocarcinoma patients. *AMIA Annu Symp Proc.* 2012;2012:1040-9.
55. Zhou T, Wang T, Garcia JG. Expression of nicotinamide phosphoribosyltransferase-influenced genes predicts recurrence-free survival in lung and breast cancers. *Sci Rep.* 2014 Aug 22;4:6107.
56. Zhu CQ, Ding K, Strumpf D, Weir BA, Meyerson M, Pennell N, Thomas RK, Naoki K, Ladd-Acosta C, Liu N, Pintelie M, Der S, Seymour L, Jurisica I, Shepherd FA, Tsao MS. Prognostic and predictive gene signature for adjuvant chemotherapy in resected non-small-cell lung cancer. *J Clin Oncol.* 2010 Oct 10;28(29):4417-24.
57. Zhu CQ, Strumpf D, Li CY, Li Q, Liu N, Der S, Shepherd FA, Tsao MS, Jurisica I. Prognostic gene expression signature for squamous cell carcinoma of lung. *Clin Cancer Res.* 2010 Oct 15;16(20):5038-47.

Response: We thank the reviewer for these comments which have led us to perform more in-depth analysis on our gene panel and made fair comparison to previously claimed prognostic gene panels. We are sorry that we were not clear enough to highlight

biological and clinical significance of our ECM gene panel in our previous manuscript. We have again added substantial insights obtained from the review of 61 studies in our discussion section and performed additional statistical computation to clearly demonstrate superior prognostic performance over traditional clinicopathological factors in all validation cohorts. The review and extra analyses in fact gave us even clearer idea how our gene panel is different from existing ones and has critical biological and clinical significance that may greatly advance our understanding of tumor microenvironment.

- We have now looked through all the 61 references that the reviewer has listed above and made a fair comparison to further highlight strengths and advantages of our gene panel over these existing prognostic signatures. For all these 61 studies, hazard ratio which represents stratifying performance together with 95% CI and p-value (often only log-rank p was stated without mentioning of HRs), gene panel, tumor staging of validation cohorts (whether early-stage specific or late stage was included), predictive value and validation cohort size are summarized (Supplementary Table 5 and Supplementary Figure 2).
- First, as opposed to these existing signatures where genes were ranked and selected solely based on their prognostic power with prior information of known survival outcomes (all except for 10 references), we were completely blinded to survival data when drawing gene signature; only existing biological knowledge was used for our gene selection, particularly for biological motives representing ECM matrix that links tumor cells and surrounding enzymes in tumor microenvironment using multiple gene ontology databases. For this, we found it remarkable that early-stage tumors of a particular kind of lung cancer, NSCLC, can be stratified with different survival outcomes based on ECM molecules alone, considering their highly variable and dynamic properties.

Because our gene selection was derived from their differential expression in NSCLC tumors, the fact that it has strikingly robust prognostic performance in stratifying early-stage tumors suggests highly orchestrated and thorough disturbances in ECM matrix in NSCLC tumors that promote disease progression (and potentially even during other lung-related inflammation for their concordant differential expression patterns in IPF (2-5), COPD (6) and CF (7), Figure 7). The possibility that these 29 specific ECM molecules may further be involved in resistance to therapeutic intervention further highlights the significance of downstream analysis of ECM mediated signaling pathways in identifying druggable targets against NSCLC tumors. Our finding of both differential expression in NSCLCs and their robust prognostic power emphasize significance of these specific ECM molecules in contributing to such diverse aspects of cancer progression.

- Because gene selection was made solely based on biological motives, this finding is significant not just in predicting survival outcomes but also in uncovering gene signature for the prioritization of therapeutic targets – particularly for stroma-targeting therapeutic approaches. Despite immense efforts, designing rational

therapy targeting CAFs, for example, has been challenged and resulted in significant toxicity due to lack of definitive biomarkers in distinguishing 'activated' CAFs from normal fibroblasts (8). Targeting CAFs in some cancers therefore have resulted in counterproductive effects which accelerated cancer progression with reduced survival, induced EMT, and suppressed anticancer immunity (9). Although several biomarkers have been identified to distinguish activated from non-activated CAFs (such as α SMA, PDPN, TNC, PDGFR α , CSPG4), these genes are concurrently expressed in other cells within tumor environment such as vascular SM cells, pericytes and mesenchymal stem cells (8). Such lack of specificity and poor understanding of tumor microenvironment at the molecular level truly necessitates specific ECM biomarkers to develop more precise and less toxic targeted therapies. In this regard, our gene panel holds great promise in uncovering novel targets for tumor microenvironment-directed therapeutic approaches – in fact, one of our ECM components, COL11A1, which encodes the α 1 chain of collagen XI, has very recently been reported as a highly specific biomarker of activated CAFs with other co-expressed genes in 13 types of cancers (8).

- The reviewer has also brought up one of the most comprehensive review articles that truly highlights serious problems associated with existing gene panels (10). We tried our best to meet the guidelines provided in this paper when designing the study, hence our gene panel was blindly ranked with biological significance using numerous gene ontology databases and not selected on the basis of some figure of merit using survival data, even from the initial gene selection. In an attempt to demonstrate the absence of any kind of bias in reporting statistical significance (as Subramanian et al. critically pointed out on the presentation of biased resubstitution statistics in prior studies), we covered a total number of 20 datasets covering both microarray and RNA-seq platforms and entire data were presented in our analyses.
- For fair comparison to our ECM gene panel, signatures selected based on biological motives from 61 studies were included for schematic demonstration (Supplementary Figure 2). Some of these signatures with biological implications include cancer-related genes such as TOP2A (11-13), PI3K-related genes (11), and VEGFA (14), which are known to be significantly associated with clinopathological factors (15, 16). Even when comparing to these signatures, our ECM gene panel demonstrates comparable and even stronger prognostic performance in greater number of validation cohorts. Nevertheless, as Subramanian et al. (10) emphasized, hazard ratio is not sufficient enough to demonstrate predictive power; thus receiver operating characteristics (ROC) curves were further obtained to demonstrate that EPPI risk score is a statistically better predictor of survival than known standard risk factors, including AJCC staging, histology, smoking status and gender, which were often missed in prior studies. Moreover, we found that prognostic performance from an average number of two independent validation datasets were included in 61 studies ('training' datasets were not considered; Supplementary Table 5), possibly owing

to non-significant statistical p-value.

Supplementary Figure 2. Comparison to previously reported gene panels with biological implications. Of existing prognostic gene panels for NSCLC tumors, those selected based on biological motives (without prior known survival information) were selected and illustrated for HR, validation sample size, number of genes and respective biological properties, which are critical factors to be considered for development of clinically applicable multigene assay for prognosis.

Comment: This list is surely not complete but illustrates impressively that it is possible to construct a gene signatures of different genes and subsequent different biological motives, e.g. proliferation, cytokines, genomic instability, etc. (Subramanian J, J Natl Cancer Inst. 2010) Also several studies addressing the tumor microenvironment in lung cancer are published:

1. Roepman P, Jassem J, Smit EF, Muley T, Niklinski J, van de Velde T, Witteveen AT, Rzyman W, Floore A, Burgers S, Giaccone G, Meister M, Dienemann H, Skrzypski M, Kozlowski M, Mooi WJ, van Zandwijk N. An immune response enriched 72-gene prognostic profile for early-stage non-small-cell lung cancer. *Clin Cancer Res.* 2009 Jan 1;15(1):284-90.
2. Navab R, Strumpf D, Bandarchi B, Zhu CQ, Pintilie M, Ramnarine VR, Ibrahimov E, Radulovich N, Leung L, Barczyk M, Panchal D, To C, Yun JJ, Der S, Shepherd FA, Jurisica I, Tsao MS. Prognostic gene-expression signature of carcinoma-associated fibroblasts in non-small cell lung cancer. *Proc Natl Acad Sci U S A.* 2011 Apr 26;108(17):7160-5.
3. Du L, Yamamoto S, Burnette BL, Huang D, Gao K, Jamshidi N, Kuo MD. Transcriptome profiling reveals novel gene expression signatures and regulating transcription factors of TGFβ-induced epithelial-to-mesenchymal transition. *Cancer Med.* 2016 Aug;5(8):1962-72.
4. Chen JL, Espinosa I, Lin AY, Liao OY, van de Rijn M, West RB. Stromal responses among common carcinomas correlated with clinicopathologic features. *Clin Cancer Res.* 2013 Sep 15;19(18):5127-35.

That the genes of the presented signatures are not completely overlapping is not surprising and not new. It is also nothing special to include only stage I and II patients, most of the studies do that, in addition they show there impact in stage I or Ia patients only, e.g. Wistuba et al. *Clin Cancer Res*, 2013; Kratz et al., 2012 *JAMA*).

Response: We thank the reviewer for bringing up this discussion regarding the ‘novelty’ of prognostic value in gene panels representing tumor microenvironment.

- The last two studies cited, unfortunately, are not relevant in this case as they concluded “there is no statistically significant clinical associations between fibroblast-associated genes in lung tumors and individual datasets of lung cancers unlike the ovarian carcinomas” (17); and neither survival analysis nor prognostic performance was done or inferred in Du et al. study (18) (only cell morphology

for mesenchymal phenotype was inferred from mRNA and protein expression using cancer cell lines).

- We believe direct comparison of our ECM gene signature to the first study is not appropriate as these 72 genes were ranked with most significant prognostic power entirely based on known prior survival data; only after using gene ontology database did they find 'subset' of these genes to be immune-response related but in fact a great proportion of this large gene panel also includes antigen binding, protein modification and degradation-related genes, not just immune-response genes (19). Even so, immune-response genes do not carry biological information in linking tumor cells and enzymes like ECM matrix and whether components of the ECM can be associated with elements of the immune system or these specific ECM molecules actually play any functional role in the immune response to cancer progression should be another research area to be investigated.
- Lastly, we appreciate the reviewer for citing Navab et al. study (20) again which reported prognostic value of 11 CAF-related genes (*ICAM-1*, *THBS2*, *MME*, *OXTR*, *PDE3B*, *CLU*, *B3GALT2*, *EVI2B*, *COL14A1*, *GAL*, and *MCTP2*). They seem to have some success in showing prognostic value of CAF-related genes, but these 11 genes were again selected with most prognostic significance from initial 46 differentially expressed CAF genes (in fact, univariate analysis results showed only 14.3% of genes having prognostic value in each probe set), and the aggregate prognostic effect of collective 46 CAF genes was neither performed nor stated. A small subset of these CAF genes from initially differentially expressed CAF genes was then selected for powerful prognostic performance and the validation cohort further includes both stage III and IV patients (supplementary table S1B in their manuscript); although staging effect was adjusted, prognostic value was not shown to early-stage specific tumors alone.

Taken together, we truly believe that the collective prognostic effect of our gene panel exclusively comprising ECM molecules which was not further manipulated to achieve better prognostic value is novel, which may uncover unique-, biologically- and therapeutically-relevant targets, particularly for stroma-directed approaches.

Comment: I miss this demonstration in the present study: The challenge how to transfer the classifier to clinical practice, i.e. on small FFPE tissue biopsies has not been addressed. Indeed there are two commercial providers for lung cancer gene expression signatures (Myriad, My Plan, lung cancer and Pervenio™ Lung RS of Life Technologies) that are applicable on paraffin embedded tissue (both not cited and discussed by the authors). That these signatures are not based on ECM is for the purpose irrelevant.

Response: We appreciate the reviewer for bringing up these two commercially available multigene assays for NSCLC. We initially chose not to mention these tests as although they are being marketed, Medicare has determined that there is still lack of evidence to warrant immediate clinical application of these assays and thus, is not reimbursing for its

use at this time (21, 22). In response to this, we now have included this in the introduction of our manuscript (line 31-37). We are sorry that we have missed out how this gene panel can be implemented for clinical application. In response to this, we have included description on potential clinical procedure using FFPE samples in the discussion section of our manuscript (line 273-281).

Comment: Finally they did not compared their model to strong clinicopathological factors (age, stage, performance status) or factors with minor prognostic impact (smoking status, sex, histology). A multivariate analysis is not sufficient and not the same. A clinical model must be compared directly to the gene signature (Subramanian J, J Natl Cancer Inst. 2010; Subramanian et al., Nature Rev Clin Oncol). Unfortunately often publically available data bases are not sufficiently annotated for this purpose. This fact does not excuse the absence of such necessary head to head comparison.

Response: The reviewer has suggested comparing with traditional clinicopathological factors. In response to this, we have performed thorough log-rank test for each known risk factors, including age, histology, gender, smoking history and tumor staging. Our EPPI model was directly compared to these parameters in all 10 validation cohorts, comprising 2,051 early-stage patients (Supplementary Figure 3). As can be seen from the figure illustration below, despite significant p-values for tumor staging, age and histology in few datasets, our gene signature still has very good prognostic performance in 8 of 10 datasets. Although EPPI risk score had significantly robust performance in the other 2 datasets, histology and tumor staging showed higher p-value in GSE30219 and TCGA, respectively. Nevertheless, our gene panel remains to be the one of the most significant factors compared to five clinicopathological factors in majority of validation cohorts.

Supplementary Figure 3. Comparison of EPPI risk score to known clinicopathological factors . As our validation cohorts are mostly complete with traditional clinicopathological factors such as age, tumor subtypes, gender, smoking status, and tumor staging, prognostic performance of EPPI gene panel was directly compared to further demonstrate strong

In summary, referring back to criteria delineated by Subramanian et al. for a clinically useful prognostic gene signature (reference has been included, discussed in lines 255-260, 340-345), we tried our very best in fulfilling most of the described standards and truly believe that these findings have been conducted in a very comprehensive and thorough manner compared to prior studies, in which biological significance in not only NSCLC but other lung-related diseases has further been implicated. Summary of our gene model in terms of critical factors to be considered in developing gene expression-based prognostic signatures (mostly compared to the factors set in the guidelines) has been illustrated in Figure 7b.

Figure 7: Biological and clinical significance of the EPPI gene signature. (a) Schematic representation of the expression patterns of the EPPI gene signature in lung cancer and other lung-related diseases. Differential expression levels in prior works were calculated using their own statistical methods from diseased lung tissues vs. matched normal lung tissues. (b) Summary of our gene panel in terms of critical factors to be considered in developing gene expression-based prognostic signatures.

For the reviewer's convenience, here are what we have added in our manuscript from the previous version of the manuscript:

1. Introduction

(line 31-37): two marketed multigene assays for NSCLC

2. Results

(line 157-165): statistically significant improvement over standard clinico-pathological risk factors (direct head to head comparison in all validation cohorts)

3. Discussion

(line 214-226): Review of 61 studies and strengths of our gene signature over existing ones

(line 228-242): Biological and clinical relevance of our gene signature in cancer research
(line 244-271, line 340-345): Fair analysis against the criteria set by Subramanian et al. for a clinically useful prognostic gene signature

(line 273-281): How FFPE samples can be used and have been proven to be feasible as clinical applicable assay in two other marketed panels

Additional figures/tables/supplementary information/reference added:

- 1. Supplementary Table 5**
- 2. Supplementary Figure 2**
- 3. Supplementary Figure 3**
- 4. Figure 7b**
- 5. Reference 7, 8, 15-22, 24**

We thank Reviewer #1 again for the time and effort putting into critically reviewing this manuscript.

Responses to Comments from Reviewer #2

Comment: My comments and criticisms were addressed by the authors. I will go through the points raised and the authors' responses in the order listed by the response letter.

Robustness of the prognostic gene list:

1. I suggested to do a single shot double derivation of the prognostic gene list, using the same method, on the basis of two completely independent cohorts (discovery, n=1118 and validation, n=1128). The authors opted for a different approach; using the discovery set only, they partitioned it into two disjoint cohorts, X and Y, calculated the gene lists for both and determined the overlap. This was repeated 10,000 times and the resulting overlap distribution was presented. Their approach has two significant advantages over my suggestion – instead of a single one shot test of overlap they have 10,000 “events”, and they kept discovery for discovery and validation – for validation, rather than mixing the two. Their approach has a disadvantage though – the different partitions are not independent, in fact any two partitions have on the average a 50% identity of patients in the respective X and Y sets. I am willing to accept their choice.

Response: We thank the reviewer for all the insightful comments which have greatly improved our manuscript previously. We chose our discovery set for its completeness for gene signatures and larger patient sample size probed with the most commonly probed platforms in gene expression studies. Thank you once again for your previous suggestion in using iterative method and random sampling and your acceptance of the change, which further confirmed that our final gene signature is not a result of any biased selection. This may be due to the fact that we were completely blinded to survival data when drawing gene signature, as opposed to existing signatures where genes were ranked and selected solely based on their prognostic power with prior information of known survival outcomes (we did a review of previously reported prognostic gene panels from 61 studies and added as Supplementary Table 5, Supplementary Figure 2 and 3);

Comment: 2. Definition of the EPPI risk score:

As I understand they have changed the risk score and their current score is much clearer than the previous one. I cannot say whether this is due to better phrasing of what was done or a more transparent procedure.

Response: We would like to show deep appreciation again for your previous suggestion on the risk scoring metrics, which has again significantly improved our manuscript.

Comment: 3. Universal threshold:

The authors took a few very significant tests in this direction. I still miss (possibly due to my own oversight) a clear “bottom line”: say I chose the optimal threshold and working point – the resulting classifier assigns patients to high/low risk groups. How good is the resulting predictor (in terms of sensitivity and specificity)? What is the risk of not giving chemotherapy to a patient who would benefit from it? This one number (evaluated at the working parameters of the predictor) is probably the most important one, together with the complementing figure (the number or fraction of patients who do not need chemo and indeed are advised not to take it? A clear statement on this would help physicians to relate to the paper's findings.

Response: We are sorry for not including sufficient explanation in our manuscript to describe the effect of changing the risk cut-off score on the sensitivity and specificity of the screening test. We have extracted full data of sensitivity (TP), specificity (1-FP), and risk score for 10 year survival from GSE50081 (181 early-stage patients) as an example to demonstrate how changing cut-off score will have an impact in patient identification:

The intersection point is the start point for an initial screening to see if there is any correlation. In this dataset, our EPPI gene panel achieved AUC of 0.786 and ‘optimal’ cut-off score at ≈22.7 (at this point, from the above figure we can see that sensitivity is ≈ 0.77 and specificity is ≈ 0.79). Nevertheless, in clinical setting, this intersection point is not always the best point; when the cut-off score is shifted to the left (<22.7), we are assuming sensitivity is more important than specificity for the assay and vice versa. In screening for early-stage NSCLC patients and potentially identifying patients with ACT benefits, we believe that achieving higher sensitivity is more critical than achieving higher specificity; higher number of patients will be identified (if tumor is present) at early-stage, and outcome (be it treatment or overall survival) greatly depends on the stage of cancer when it is diagnosed, especially in NSCLC tumors. If patients are put at substantial risk by receiving false positive results, it may be worth compromising sensitivity – in which we believe is less significant case in early-stage NSCLC than detecting more cases of cancer at early-stage for greater benefits. We now have added this discussion in our manuscript (line 347-360).

We thank Reviewer #2 again for the valuable insights and suggestions.

Responses to Comments from Reviewer #3

Comment: The authors have addressed my concerns very well.

Response: We thank this reviewer for the time and effort in reviewing our manuscript.

References

1. Flaherty KT, Infante JR, Daud A, Gonzalez R, Kefford RF, Sosman J, et al. Combined BRAF and MEK inhibition in melanoma with BRAF V600 mutations. *N Engl J Med.* 2012;367(18):1694-703.
2. Meltzer EB, Barry WT, D'Amico TA, Davis RD, Lin SS, Onaitis MW, et al. Bayesian probit regression model for the diagnosis of pulmonary fibrosis: proof-of-principle. *BMC medical genomics.* 2011;4:70.

3. Yang IV, Coldren CD, Leach SM, Seibold MA, Murphy E, Lin J, et al. Expression of cilium-associated genes defines novel molecular subtypes of idiopathic pulmonary fibrosis. *Thorax*. 2013;68(12):1114-21.
4. Konishi K, Gibson KF, Lindell KO, Richards TJ, Zhang Y, Dhir R, et al. Gene expression profiles of acute exacerbations of idiopathic pulmonary fibrosis. *Am J Respir Crit Care Med*. 2009;180(2):167-75.
5. DePianto DJ, Chandriani S, Abbas AR, Jia G, N'Diaye EN, Caplazi P, et al. Heterogeneous gene expression signatures correspond to distinct lung pathologies and biomarkers of disease severity in idiopathic pulmonary fibrosis. *Thorax*. 2015;70(1):48-56.
6. Bhattacharya S, Srisuma S, Demeo DL, Shapiro SD, Bueno R, Silverman EK, et al. Molecular biomarkers for quantitative and discrete COPD phenotypes. *American journal of respiratory cell and molecular biology*. 2009;40(3):359-67.
7. Clarke LA, Sousa L, Barreto C, Amaral MD. Changes in transcriptome of native nasal epithelium expressing F508del-CFTR and intersecting data from comparable studies. *Respiratory research*. 2013;14:38.
8. Jia D, Liu Z, Deng N, Tan TZ, Huang RY, Taylor-Harding B, et al. A COL11A1-correlated pan-cancer gene signature of activated fibroblasts for the prioritization of therapeutic targets. *Cancer Lett*. 2016;382(2):203-14.
9. Ozdemir BC, Pentcheva-Hoang T, Carstens JL, Zheng X, Wu CC, Simpson TR, et al. Depletion of carcinoma-associated fibroblasts and fibrosis induces immunosuppression and accelerates pancreas cancer with reduced survival. *Cancer Cell*. 2014;25(6):719-34.
10. Subramanian J, Simon R. Gene expression-based prognostic signatures in lung cancer: ready for clinical use? *J Natl Cancer Inst*. 2010;102(7):464-74.
11. Rajski M, Saaf A, Buess M. BMP2 response pattern in human lung fibroblasts predicts outcome in lung adenocarcinomas. *BMC medical genomics*. 2015;8:16.
12. Bueno R, Hughes E, Wagner S, Gutin AS, Lanchbury JS, Zheng Y, et al. Validation of a molecular and pathological model for five-year mortality risk in patients with early stage lung adenocarcinoma. *Journal of thoracic oncology : official publication of the International Association for the Study of Lung Cancer*. 2015;10(1):67-73.
13. Wistuba, II, Behrens C, Lombardi F, Wagner S, Fujimoto J, Raso MG, et al. Validation of a proliferation-based expression signature as prognostic marker in early stage lung adenocarcinoma. *Clin Cancer Res*. 2013;19(22):6261-71.
14. Buffa FM, Harris AL, West CM, Miller CJ. Large meta-analysis of multiple cancers reveals a common, compact and highly prognostic hypoxia metagene. *Br J Cancer*. 2010;102(2):428-35.
15. Martins SF, Garcia EA, Luz MA, Pardal F, Rodrigues M, Filho AL. Clinicopathological correlation and prognostic significance of VEGF-A, VEGF-C, VEGFR-2 and VEGFR-3 expression in colorectal cancer. *Cancer Genomics Proteomics*. 2013;10(2):55-67.
16. Sullivan GF, Amenta PS, Villanueva JD, Alvarez CJ, Yang JM, Hait WN. The expression of drug resistance gene products during the progression of human prostate cancer. *Clin Cancer Res*. 1998;4(6):1393-403.
17. Chen JL-Y, Espinosa I, Lin AY, Liao OY-W, van de Rijn M, West RB. Stromal Responses among Common Carcinomas Correlated with Clinicopathologic Features. *Clin Cancer Res*. 2013;19(18):5127-35.

18. Du L, Yamamoto S, Burnette BL, Huang D, Gao K, Jamshidi N, et al. Transcriptome profiling reveals novel gene expression signatures and regulating transcription factors of TGFbeta-induced epithelial-to-mesenchymal transition. *Cancer medicine*. 2016;5(8):1962-72.
19. Roepman P, Jassem J, Smit EF, Muley T, Niklinski J, van de Velde T, et al. An immune response enriched 72-gene prognostic profile for early-stage non-small-cell lung cancer. *Clin Cancer Res*. 2009;15(1):284-90.
20. Navab R, Strumpf D, Bandarchi B, Zhu CQ, Pintilie M, Ramnarine VR, et al. Prognostic gene-expression signature of carcinoma-associated fibroblasts in non-small cell lung cancer. *Proceedings of the National Academy of Sciences of the United States of America*. 2011;108(17):7160-5.
21. Zheng Y, Bueno R. Commercially available prognostic molecular models in early-stage lung cancer: a review of the Pervenio Lung RS and Myriad myPlan Lung Cancer tests. *Expert review of molecular diagnostics*. 2015;15(5):589-96.
22. National Human Genome Research Institute. Reimbursement Models to Promote Evidence Generation and Innovation for Genomic Tests. . Available from: <http://www.genome.gov/27552210>.